# The midbody interactome reveals unexpected roles for PP1 phosphatases in cytokinesis

Luisa Capalbo[1], Zuni I. Bassi [1], Marco Geymonat[2], Sofia Todesca [1], Liviu Copoiu [1,10], Anton J. Enright[1], Giuliano Callaini[3], Maria Giovanna Riparbelli[4], Lu Yu[5,11], Jyoti S. Choudhary [5,11], Enrico Ferrero [2,6,12], Sally Wheatley [7], Max E. Douglas[8,13], Masanori Mishima[8,9] & Pier Paolo D'Avino[1]*

The midbody is an organelle assembled at the intercellular bridge between the two daughter cells at the end of mitosis. It controls the final separation of the daughter cells and has been involved in cell fate, polarity, tissue organization, and cilium and lumen formation. Here, we report the characterization of the intricate midbody protein-protein interaction network (interactome), which identifies many previously unknown interactions and provides an extremely valuable resource for dissecting the multiple roles of the midbody. Initial analysis of this interactome revealed that PP1β-MYPT1 phosphatase regulates microtubule dynamics in late cytokinesis and de-phosphorylates the kinesin component MKLP1/KIF23 of the centralspindlin complex. This de-phosphorylation antagonizes Aurora B kinase to modify the functions and interactions of centralspindlin in late cytokinesis. Our findings expand the repertoire of PP1 functions during mitosis and indicate that spatiotemporal changes in the distribution of kinases and counteracting phosphatases finely tune the activity of cytokinesis proteins.

[1] Department of Pathology, University of Cambridge, Tennis Court Road, Cambridge CB2 1QP, UK. [2] Department of Genetics, University of Cambridge, Downing Street, Cambridge CB2 3EH, UK. [3] Department of Medical Biotechnologies, University of Siena, Via A. Moro 4, 53100 Siena, Italy. [4] Department of Life Sciences, University of Siena, Via A. Moro 4, 53100 Siena, Italy. [5] The Wellcome Trust Sanger Institute, Wellcome Genome Campus, Hinxton CB10 1SA, UK. [6] Cambridge Systems Biology Centre, University of Cambridge, Tennis Court Road, Cambridge CB2 1QR, UK. [7] School of Life Sciences, University of Nottingham, Nottingham NG7 2UH, UK. [8] Wellcome Trust/Cancer Research UK Gurdon Institute, Cambridge CB2 1QN, UK. [9] Centre for Mechanochemical Cell Biology and Division of Biomedical Sciences, Warwick Medical School, University of Warwick, Coventry CV4 7AL, UK. [10]Present address: Department of Biochemistry, University of Cambridge, Cambridge, UK. [11]Present address: The Institute of Cancer Research, 123 Old Brompton Road, London SW7 3RP, UK. [12]Present address: Autoimmunity, Transplantation and Inflammation Bioninfomatics, Novartis Institutes for BioMedical Research, 4056 Basel, Switzerland. [13]Present address: Chromosome Replication Laboratory, The Francis Crick Institute, 1 Midland Road, London NW1 1AT, UK. *email: ppd21@cam.ac.uk

Growth, development, and reproduction in multicellular organisms depend on the faithful segregation of genomic and cytoplasmic material that occurs during cell division. Errors during this process are responsible for many human diseases, including cancer. In the final step of cell division, the mother cell divides into two daughter cells during the process of cytokinesis. This major cell shape change requires the assembly and coordinated activity of two cytoskeletal structures: the actomyosin contractile ring, which assembles at the equatorial cortex and drives the ingression of the cleavage furrow; and the central spindle, an array of anti-parallel and interdigitating microtubules, which is essential for positioning the cleavage furrow, keeping the dividing genomes apart, and for the final separation, i.e., abscission, of the daughter cells[1]. The contractile ring and the central spindle are composed of several proteins and protein complexes that act as structural and regulatory factors that control the formation, dynamics and stability of these cytoskeletal structures throughout cytokinesis[1,2]. Like in many other processes during cell division, the functions and interactions of these proteins are often regulated by reversible posttranslational modifications, including phosphorylation/dephosphorylation mostly mediated by serine/threonine kinases and their counteracting phosphatases[3]. During furrow ingression, the contractile ring compacts the central spindle and, after completion of furrow ingression, the two daughter cells remain connected by an intercellular bridge, which contains at its center an organelle, the midbody, composed of a multitude of proteins that have diverse functions. Some midbody proteins are former components of the contractile ring and central spindle, while others are specifically recruited during the slow midbody maturation process that ultimately leads to the abscission of the two daughter cells[4,5]. All these proteins are arranged in a very precise and stereotyped spatial pattern along the midbody[6], which can be divided in approximately three major regions: the midbody ring, containing mostly former contractile ring components like Anillin and Citron kinase; the midbody central core marked by central spindle proteins such as the centralspindlin complex; and the midbody arms that flank the midbody core and where the chromosomal passenger complex (CPC) and the kinesin KIF20A accumulate[4]. The proper localization, regulation and interactions of all these proteins are essential for the execution of abscission and for preventing incorrect genome segregation[5]. Furthermore, recent evidence indicates that the midbody is also involved in many other processes besides cell division, including cell fate, pluripotency, apical-basal polarity, tissue organization, and cilium and lumen formation[7,8]. Therefore, the characterization of the intricate midbody protein interaction networks (i.e., interactome) and their regulation is essential for understanding how this organelle executes its multiple functions.

In this study, we report the characterization of the midbody interactome identified by affinity purifications coupled with mass spectrometry (AP-MS) of ten key midbody components. This valuable resource provides a molecular blueprint of the intricate connections amongst midbody components that will be pivotal in dissecting the multiple functions of this organelle. In support of this, our initial analysis of the midbody interactome already revealed a plethora of previously unidentified interactions and highlighted a role of the PP1β-MYPT1 phosphatase in regulating the dynamics of central spindle microtubules by antagonizing Aurora B phosphorylation of the centralspindlin component MKLP1 in late cytokinesis.

## Results

### CIT-K interactions increase specifically during cytokinesis.
Citron kinase (CIT-K) is a contractile ring component that acts as a major midbody organizer by interacting with several midbody components, including the CPC and centralspindlin, and by maintaining their correct localization and orderly arrangement[9,10]. As a first step toward the characterization of the midbody interactome, we used a human HeLa cell line stably expressing CIT-K tagged with GFP[11] to identify the CIT-K interactomes at different cell cycle stages—S phase, metaphase, and telophase—by AP-MS (Fig. 1a, b and Supplementary Data 1). We found that the number of CIT-K interactors consistently increased in telophase in three separate replicates, confirming the important role of this kinase in cytokinesis (Fig. 1b). Notably, only 62 proteins, including the bait CIT-K and two of its known partners, the contractile ring component Anillin and the kinesin KIF14[12,13], were common to all three mitotic stages (Fig. 1b and Supplementary Data 1), indicating that our AP-MS methodology identifies specific interactions and generates little noise.

To assess whether CIT-K was required for recruiting some of these interactors to the midbody, we used SILAC-based quantitative MS to characterize and compare the proteomes of midbodies purified from telophase HeLa cells treated with either CIT-K or control siRNAs (Fig. 1c, Supplementary Fig. 1, and Supplementary Data 2). Only minor differences in the levels of a few midbody proteins were identified, including Filamin B, the kinesin KIFC1, Aurora A kinase and its interactor TPX2 (Fig. 1d, Supplementary Table 1 and Supplementary Data 2). Although some of these differences were significant and validated by western blot (Fig. 1e and Supplementary Table 1), overall our results did not indicate a major role for CIT-K in recruiting midbody proteins and reinforced the evidence that CIT-K has a very specific function in the organization of this organelle.

Finally, it is important to point out that our midbody proteome contains a significant higher number of proteins than a previous study[14]. This most likely reflects the considerable advancements in MS technology in recent years rather than a difference in the midbody purification protocols.

### The midbody interactome has common and specific networks.
To further our knowledge of the interaction networks within the midbody, we expanded our AP-MS experiments to include nine additional baits, all proteins that are well known to play key roles in midbody assembly and cytokinesis and display specific and distinct localization patterns (Table 1). AP-MS analysis of the interactions of these ten baits in telophase revealed a complex midbody interactome comprising almost 3000 proteins (Supplementary Data 3, 4), which included the majority of midbody proteome components and showed a Gene Ontology (GO) enrichment profile very similar to the midbody proteome characterized in our SILAC experiments (Fig. 2 and Supplementary Data 4, 5). The overlap and similarity between the two datasets is highly significant considering that they were obtained using two completely different experimental procedures (see Methods). The midbody interactome contains complex networks shared by several baits as well as networks specific for each bait or for just a few baits (Fig. 3a–d). Interestingly, even proteins strictly related, like the two ESCRT-III paralogs CHMP4B and CHMP4C, showed distinct specific networks (Fig. 3a). Analysis of the interactome networks further confirmed the specificity and selectivity of our AP-MS methodology. For example, the contractile ring component Anillin presents a specific interaction network that includes the vast majority of septin proteins (Fig. 3b), which are known to be recruited by Anillin to midbody[15,16]. Similarly, the mitotic kinase Polo-like kinase 1 (Plk1) was only identified with the bait PRC1 (Fig. 3d), which directly binds to and recruits Plk1 to the central spindle and midbody in both human and Drosophila cells[17,18].

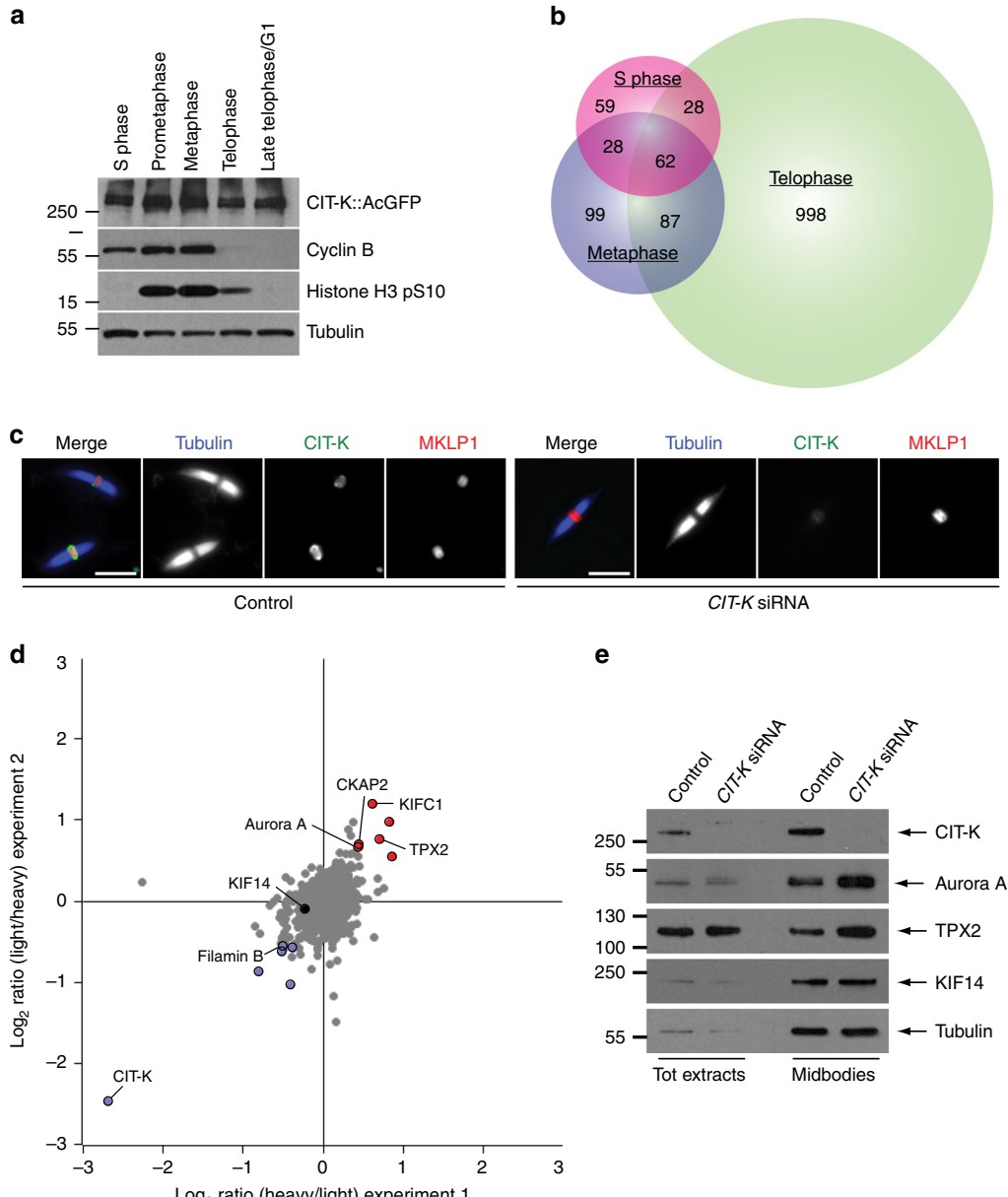

**Fig. 1** CIT-K specifically interacts with a multitude of proteins in cytokinesis. **a** Western blot analysis of protein extracts from HeLa cells stably expressing a CIT-K::AcGFP transgene synchronized at different stages of the cell cycle. The blots were probed with antibodies against the proteins indicated to the right. Numbers on the left indicate the size, in kDa, of the protein ladder. **b** Proportional Venn diagram showing the number of proteins identified at each cell cycle stage by AP-MS using CIT-K::AcGFP as bait. **c** Midbodies purified from HeLa S3 cells treated with siRNAs directed against either a random sequence (control) or *CIT-K* were fixed and stained to detect tubulin, CIT-K, and MKLP1. Scale bars, 5 μm. **d** Logarithmic normalized protein ratios from two independent SILAC experiments were plotted against each other. Each point represents a single protein identified. Gray dots correspond to proteins that did not show any significant difference in abundance between control and CIT-K siRNA midbodies. Red and blue dots represent proteins that were either significantly enriched or less abundant after CIT-K depletion in both biological replicates ($p$ value < 0.01; significance B test corrected by Benjamini-Hochberg method). **e** Western blot analysis of total protein extracts and midbodies purified from telophase HeLa S3 cells treated with siRNAs directed against either a random sequence (control) or *CIT-K*. The blots were probed with antibodies against the proteins indicated to the right. Numbers on the left indicate the size, in kDa, of the protein ladder. Source data for Fig. 1a and e are provided as a Source Data file

Our midbody interactome revealed a plethora of interactions that can lead to the discovery of important structural and regulatory networks present within this organelle. For example, in the network specifically shared by CIT-K and its partner KIF14, we identified the cyclin-dependent kinase 1 (Cdk1) and the microtubule depolymerizing kinesin KIF2C/MCAK (Fig. 3c), both also found in the midbody proteome (Supplementary Data 2 and 4). KIF2C regulates microtubule dynamics during mitotic spindle assembly[19], but it has not

been implicated in cytokinesis. Cdk1, in complex with cyclin B, is well known to promote mitotic entry and to regulate multiple mitotic events until anaphase, when most of the complex is inactivated through degradation of cyclin B. However, a pool of Cdk1/cyclin B has been described to accumulate at the midbody where it appears to promote abscission[20]. Our data not only indicate a potential role for KIF2C in cytokinesis and confirm the presence and function of Cdk1/cyclin B at the midbody, but also suggest that the

**Table 1 Baits used in the AP-MS experiments for the characterization of the midbody interactome**

| Name | Function | Localization | Tag and cell line reference |
|---|---|---|---|
| Anillin | Actomyosin binding protein; contractile ring scaffolding | Cleavage furrow, midbody ring and secondary constriction sites | GFP[48] |
| Aurora B | Serine/threonine kinase; CPC component, controls furrow ingression, central spindle formation and abscission | Cleavage furrow, central spindle, midbody arms | GFP (this study) |
| CHMP4B | ESCRT-III protein; required for abscission | Midbody core and abscission site | GFP[49] |
| CHMP4C | ESCRT-III protein; required for abscission | Central spindle, midbody arms, midbody core and abscission site | Flag[26] |
| Citron kinase (CIT-K) | Serine/threonine kinase; required for midbody assembly, organization and maturation | Cleavage furrow, central spindle, midbody ring | AcGFP[11] |
| Ect2 | Rho GEF; activates RhoA to promote contractile ring assembly and constriction | Cleavage furrow, central spindle, midbody core | AcGFP[50] |
| KIF14 | Kinesin, CIT-K partner; required for midbody assembly, organization and maturation | Cleavage furrow, central spindle, midbody ring | GFP[51] |
| KIF20A/MKLP2 | Kinesin; required for central spindle formation and CPC translocation | Central spindle, midbody arms | GFP[51] |
| KIF23/MKLP1 | Kinesin; centralspindlin component, required for furrow ingression, central spindle and midbody formation | Central spindle, midbody core | GFP[52] |
| PRC1 | Microtubule associated protein, required for central spindle and midbody formation | Central spindle, internal midbody arms and core | GFP[53] |

association with CIT-K and KIF14 might be important for their localization and/or function.

**PP1β-MYPT1 controls microtubule dynamics in late cytokinesis.** Cell division is regulated by posttranslational modifications, including phosphorylation mostly mediated by serine/threonine kinases and counteracting phosphatases. Although most kinases involved in cytokinesis are known, the identity and function of their opposing phosphatases is just emerging[4,21]. To address this nescience, we generated a midbody interactome serine/threonine phosphorylation sub-network by extracting from the entire interactome dataset proteins whose full names (Uniprot field: protein names) contained the terms kinase and phosphatase but excluded those containing tyrosine (Fig. 3d). The most frequent and abundant phosphatases belong to the PP1 family (Supplementary Table 2), and the top scores include the three PP1 catalytic subunits—α, β, and γ—and the PPP1R12A regulatory subunit, also known as myosin phosphatase target subunit 1, MYPT1. PP1γ was described to accumulate at the cleavage site[22] and we found that all four PP1 phosphatases localized to the midbody ring in late cytokinesis, and PP1β and MYPT1 also accumulated at the cleavage furrow in early telophase (Fig. 4a–d). The signals detected by these antibodies are specific because they were strongly reduced after siRNA treatments in both immuno-fluorescence and western blot analyses (Fig. 4a–e). We then investigated if siRNA-mediated depletion of these phosphatases caused cytokinesis failure. siRNA of *PP1β* and *MYPT1* caused the highest increases in multinucleation (a readout of cytokinesis failure), 4.2- and 7.2-fold, respectively (Fig. 4g, h). PP1α depletion did not result in an increase of multinucleated cells and only a very modest increase (1.6-fold) was observed after PP1γ siRNA (Fig. 4h). However, combined depletion of these two closely related catalytic subunits resulted in a 2.8-fold increase in multi-nucleated cells (Fig. 4h), suggesting that they could act redundantly and/or synergistically in cytokinesis. In sum, our results indicated that, of all four phosphatases, PP1β and MYPT1 were the two most strongly required for cytokinesis (Fig. 4e, g, h), which is consistent with the evidence that MYPT1 is a known PP1β regulatory subunit[23]. MYPT1 was reported to antagonize Plk1 during mitotic spindle assembly and to be required for cytokinesis[24], but its exact role in cytokinesis was not investigated,

probably assuming that it was required to de-phosphorylate the myosin regulatory light chain (MRLC) at the contractile ring. We found that, indeed, the levels of both mono(pS19)- and di(pT18 pS19)-phosphorylated MRLC levels were elevated in MYPT1 depleted cells (Fig. 5a, b), which had also an abnormal cytoskeleton and numerous cortical blebs (Fig. 4f). However, mitotic exit was not affected after MYPT1 siRNA, as cyclin B levels dropped in anaphase and dephosphorylation of two phospho-epitopes, PRC1 pT481[25] and tri-phospho CHMP4C[26,27], known to occur upon mitotic exit, was not affected (Fig. 5b). *MYPT1* siRNA cells could successfully complete furrowing, although the central spindle appeared longer and bent upwards in late cytokinesis (Fig. 5a and Supplementary Movies 1–4). Time-lapse analysis of chromosome and microtubule dynamics during cell division revealed that *MYPT1* siRNA caused abnormal cortical contractility that did not prevent furrow formation and ingression, albeit furrowing was faster than in control cells (Fig. 6a–c, and Supplementary Movies 5–8), likely because of hyper-phosphorylated MRLC. Notably, after completion of furrow ingression, *MYPT1* siRNA cells failed to maintain a robust central spindle, which became very thin, bent and long, and sometime snapped (Fig. 6a–h, and Supplementary Movie 6). Consistent with these phenotypes, in the majority of *MYPT1* siRNA cells abscission either failed or did not occur during the period of filming (Fig. 6b and Supplementary Movie 7). Even when *MYPT1* siRNA cells could successfully separate, abscission was significantly delayed (Fig. 6c). Furthermore, in MYPT1 depleted cells, the midbody was not properly assembled as many of its components were stretched along the central spindle and lost their precise arrangement: Aurora B kinase spread from the midbody arms into the midbody core (Fig. 6d), the kinesin MKLP1 and the microtubule bundling protein PRC1 failed to localize as two juxtaposed disks (Figs. 6e–g and 7a), and CIT-K assembled into misshapen rings that often collapsed (Fig. 6e, h). Electron microscopy analysis showed that *MYPT1* siRNA midbodies contained fewer microtubules and an abnormal midbody matrix compared with control cells (Fig. 6i). These central spindle and midbody defects are not linked to abnormal cortical contractions or adhesion problems because they were also observed in less adherent HeLa S3 cells, which do not form cortical blebs after MYPT1 depletion (Supplementary Fig. 2). Furthermore, very similar results were obtained in immortalized,

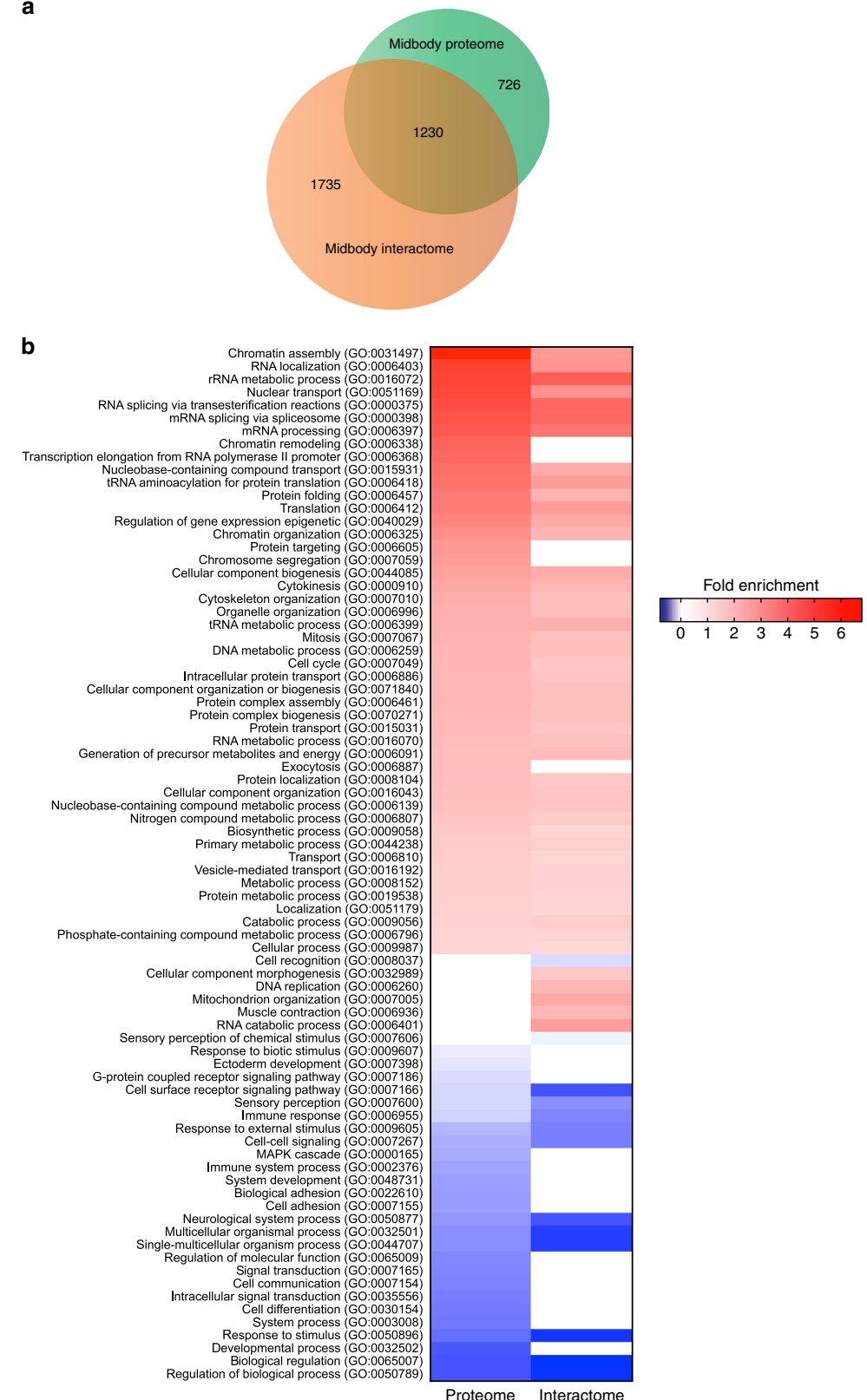

**Fig. 2** The midbody proteome and interactome share many proteins and have similar GO enrichment profiles. **a** Proportional Venn diagram showing the number of proteins identified in the midbody proteome and interactome. The majority of midbody proteome proteins (62.9%) are contained in the midbody interactome. **b** Heat map showing the GO annotation enrichment profiles of the midbody proteome and of the midbody interactome. GO enrichment profiles were analyzed using PANTHER under the category GO-slim biological process. Overrepresented GO terms are shown in shades of red while underrepresented GO terms are shown in shades of blue, according to their fold enrichment as indicated in the color scale bar at the right. Only Bonferroni-corrected results for $p < 0.05$ were considered (see Supplementary Data 5)

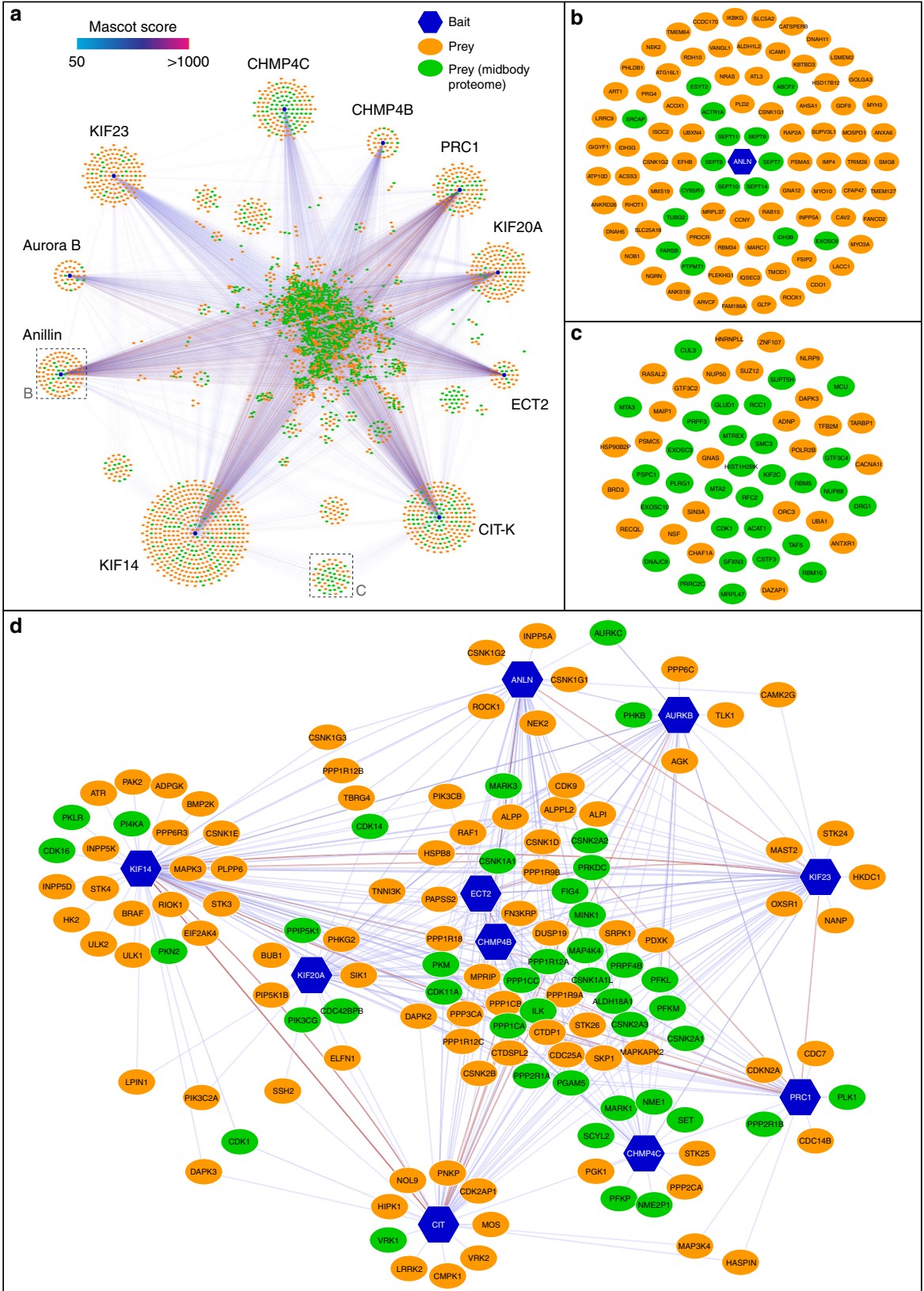

**Fig. 3** The midbody interactome comprises common and specific networks. **a** Diagram illustrating the entire midbody interactome. Baits are indicated with blue hexagons, while preys are represented as ovals, either in green, if they were also found in the midbody proteome, or in orange. The edges connecting the network nodes are colored according to their Mascot scores as indicated in the color scale bar at the top left. Preys shared by multiple baits are clustered in the center. **b** Enlargement of the Anillin-specific sub-network shown in the corresponding inset in **a**. **c** Enlargement of the baits shared specifically by CIT-K and KIF4 shown in the corresponding inset in **a**. **d** Diagram representing the phosphorylation sub-network. All nodes are labeled with their primary gene names according to the UniProt database (https://www.uniprot.org)

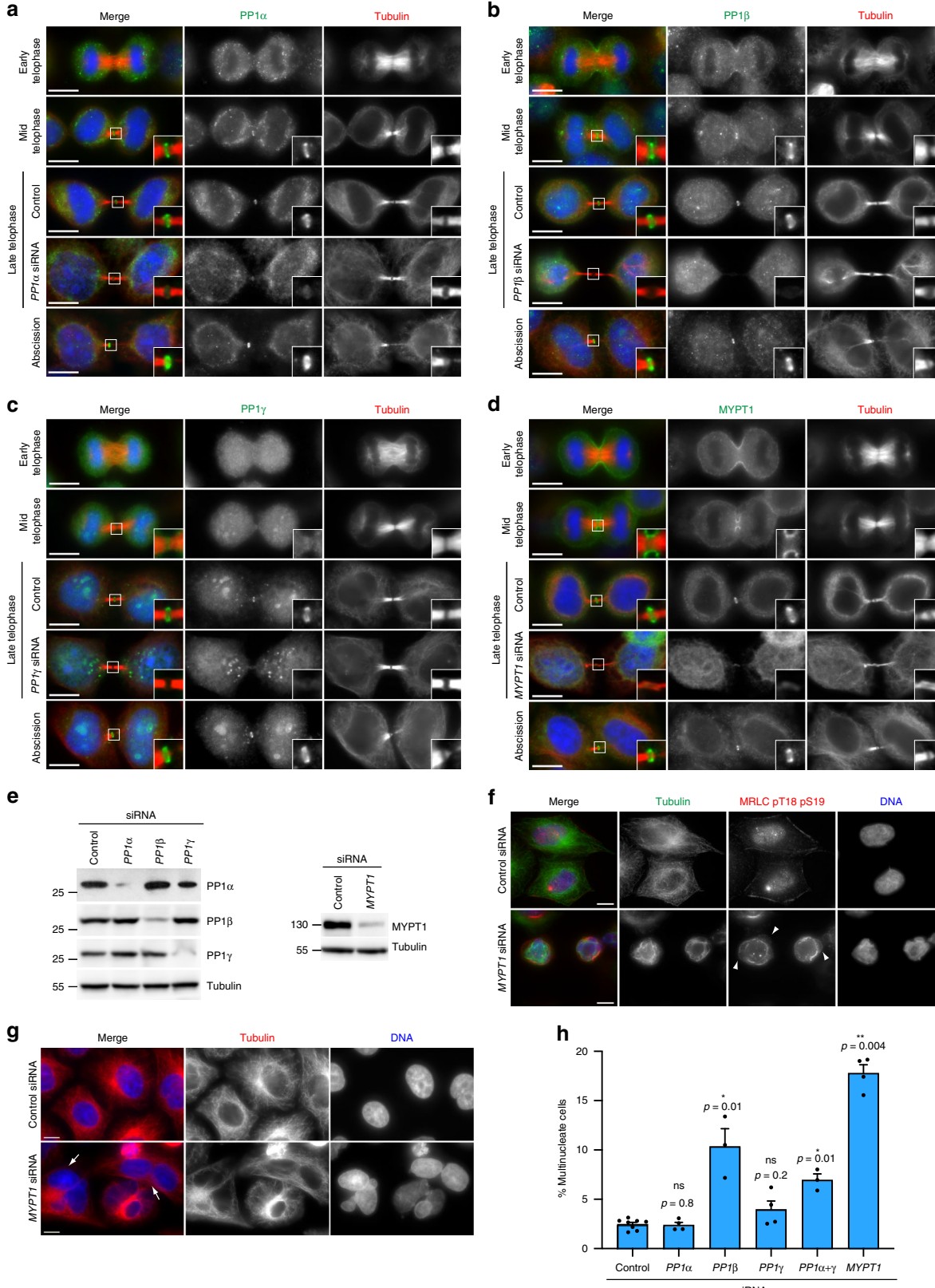

non-transformed RPE-1 cells (Supplementary Fig. 3), indicating a general requirement for MYPT1 in cytokinesis in different cell types. Finally, almost identical phenotypes were observed after *PP1β* siRNA (Fig. 4b and Supplementary Fig. 4), further supporting that MYPT1 is acting as a regulatory subunit for PP1β in late cytokinesis.

**PP1β dephosphorylates the centralspindlin component MKLP1.** Central spindle assembly depends on various microtubule associated proteins (MAPs)[2], including two key protein complexes: centralspindlin, a hetero-tetramer composed of two MKLP1 and two RacGAP1 subunits, and the PRC1-KIF4A complex. These MAPs have been shown to interact and cooperate

**Fig. 4** PP1 phosphatases localize to the midbody and depletion of PP1β and MYPT1 causes cytokinesis failure. **a–d** HeLa cells were fixed and stained to detect to detect DNA (blue in the merged panels), tubulin, and PP1α (**a**), PP1β (**b**), PP1γ (**c**), and MYPT1 (**d**). For RNAi depletions, HeLa cells were treated with siRNAs directed against each of the three PP1 catalytic subunits or MYPT1 and after 48 h were fixed and stained to detect the same epitopes as described above. DNA condensation and shape and thickness of microtubule bundles at the intercellular bridge were used as criteria to stage telophase cells. Insets show a 3× magnification of the midbody. Scale bars, 10 μm. **e** HeLa Kyoto cells were treated with siRNAs directed against either a random sequence (control) or each of the three PP1 catalytic subunits (left) or MYPT1 (right) and after 48 h proteins were extracted and analyzed by western blot to detect the indicated proteins. The numbers on the left indicate the sizes in kDa of the molecular mass marker. **f** HeLa cells were treated with siRNAs directed against either a random sequence (control) or MYPT1 and after 48 h were fixed and stained to detect DNA, tubulin, and di-phosphorylated MRLC. Note that MYPT1 siRNA cells show abnormal cell and nuclear shape, cortical blebs (arrowheads) and disorganized microtubule and actomyosin cytoskeletal filaments. Scale bars, 10 μm. **g** HeLa cells were treated with siRNAs directed against either a random sequence (control) or MYPT1 and after 48 h were fixed and stained to detect DNA and tubulin. The arrows indicate multinucleate cells. Scale bars, 10 μm. **h** Quantification of multinucleate cells obtained after siRNA of the three PP1 catalytic subunits or MYPT1. More than 500 cells were counted in $n \geq 3$ independent experiments. Bars indicate standard errors. *$p < 0.05$, **$p < 0.01$ (Mann–Whitney U test). Source data for Fig. 4e and h are provided as a Source Data file

to increase the robustness of the central spindle[28]. To understand the molecular mechanisms underpinning the phenotypes observed after MYPT1-PP1β depletion, we investigated whether centralspindlin could be one of the substrates of this phosphatase. Centralspindlin clustering at the central spindle midzone is necessary for its localization and function and requires phosphorylation of the evolutionarily conserved MKLP1 S708 residue by Aurora B[29]. MYPT1 depletion caused a significant increase in MKLP1 S708 phosphorylation at the midbody (Fig. 7a, b), reduced the association of this kinesin with its RacGAP1 partner and almost completely abolished its interaction with PRC1 and PP1β, but only mildly affected the association with CIT-K (Fig. 7c). MKLP1 contains a highly conserved VQF motif 80 amino acids downstream of S708 (aa 786–788; Fig. 7d and Supplementary Fig. 5a) that partially matches the RVxF consensus binding site for PP1 catalytic subunits[30]. We found that full length MKLP1 and PP1β interacted when co-expressed in yeast (Fig. 7e) and that the MKLP1 C-terminal region (aa 620–858) purified from bacteria was also able to pull down PP1β in vitro (Supplementary Fig. 5b-d), indicating that PP1β directly binds to the MKLP1 C-terminus. MKLP1$_{620-858}$ was dephosphorylated at S708 by PP1β in vitro (Fig. 7f, g) and when the VQF residues were mutated to AQA the binding of MKLP1 to PP1β was reduced (Fig. 7e and Supplementary Fig. 5b-d) and MKLP1$_{620-858}$ dephosphorylation by PP1β in vitro was less efficient (Fig. 7f, g). To assess the role of MKLP1 dephosphorylation by PP1β in vivo, we generated cell lines stably expressing GFP-tagged versions of either wild type MKLP1 or of the mutant containing the AQA mutation at residues 786–788. Silencing MKLP1 by using an siRNA directed against its 3′UTR that is absent in the GFP-tagged transgenes severely impaired central spindle assembly and cleavage furrow ingression (Fig. 7h, top panels). The very few MKLP1 siRNA cells that managed to complete furrowing had very thin central spindles and abnormal PRC1 localization (Fig. 7h), resembling MYPT1 siRNA cells (Fig. 6). The MKLP1$^{AQA}$ mutant rescued cytokinesis failure after depletion of endogenous MKLP1 much less efficiently than the wild-type counterpart (Fig. 7h, i). Importantly, MKLP1$^{AQA}$ could successfully rescue the initial stages of cytokinesis, but in late telophase central spindles appeared thin and PRC1 spread along central spindle microtubules (Fig. 7h, bottom panels), again similar to MYPT1 siRNA cells (Fig. 6). Together, these results indicate that PP1β dephosphorylates MKLP1 at S708 in late cytokinesis via association with the VQF motif and that this dephosphorylation is important for MKLP1 function in late cytokinesis.

## Discussion
Our characterization of the midbody interactome and proteome represents a significant advance in understanding the complex and intricate protein–protein interactions of this organelle. Our interactome is derived from experimental data and provides a much more realistic and accurate picture than a previous bioinformatics study[31]. The overlap and highly similar GO enrichment profiles of the interactome and proteome datasets (Fig. 2) strongly support the validity of our approach and methodology. As expected, both datasets are enriched in proteins involved in mitosis and cytokinesis, but they also show a significant enrichment in proteins involved in chromatin assembly and mRNA processing and translation (Fig. 2 and Supplementary Data 4, 5). Although unpredicted, these findings are consistent with the identification of histones at the midbody[32] and the evidence that the RNA-binding protein ATX-2 is involved in posttranscriptional regulation of PAR-5 levels at the midbody[33]. They also highlight the possibility that the midbody may function as a translational hub, which could indicate a mechanism by which asymmetric inheritance of the midbody imparts genetic information in cell fate and carcinogenesis[34,35]. The identification of common and specific networks of midbody proteins could serve to dissect main regulatory mechanisms and pathways for midbody function as well as to identify specific roles for each of the ten baits used in our study. Together, these specific and common interaction networks will undoubtedly provide an extremely valuable resource for understanding the emerging multifaceted biological roles of this organelle. However, our interactome analysis is limited to one cell type and it is possible that different proteins and protein–protein interactions exist in midbodies of different cell types. Nevertheless, our study provides a molecular blueprint of the interaction networks in the midbody, which can serve to identify major nodes, hubs and pathways that may facilitate the analysis and comparison of midbodies in other cellular and developmental contexts.

The most abundant and frequent phosphatases identified in our midbody interactome are members of the PP1 family (Fig. 3d and Supplementary Table 2). This was somehow unexpected as only PP2A phosphatases had been previously implicated in the regulation of cytokinesis[36,37] and just very recently a role in abscission was described for PP1γ and its targeting co-factor RIF1[38], which was also identified in both our midbody interactome and proteome (Supplementary Data 4). Therefore, our results expand the repertoire of PP1 functions during mitotic exit and indicate that MYPT1-PP1β is required to regulate the pace of cleavage furrow ingression and to form strong and stable central spindles and midbodies in late cytokinesis (Fig. 6a–c). Cytokinesis failure after MYPT1 siRNA occurs predominantly at a late stage, after completion of furrow ingression (Fig. 6b), highlighting an unanticipated role of MYPT1 in this phase of cell division. Our results indicate that MYPT1-PP1β regulates the dynamics of the two major cytokinetic structures, the actomyosin contractile ring and the central spindle, by de-phosphorylating different

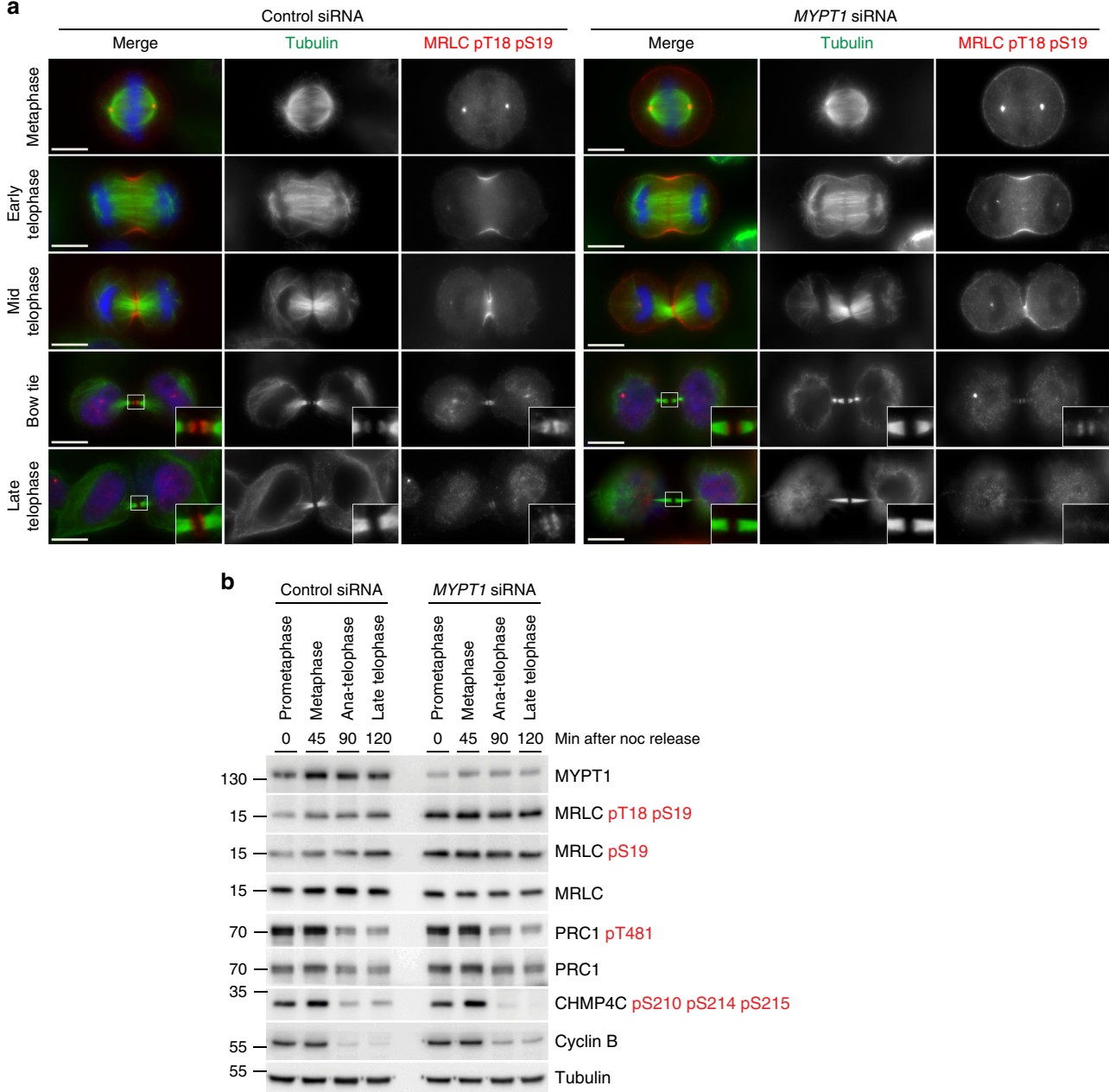

**Fig. 5** *MYPT1* siRNA increases the levels of phosphorylated MRLC, but does not impair furrowing and dephosphorylation during mitotic exit. **a** HeLa cells were treated with siRNAs directed against either a random sequence (control) or *MYPT1* and after 48 h were fixed and stained to detect DNA (blue in the merged panels), tubulin, and di-phosphorylated MRLC pT18 pS19. DNA condensation and the shape and thickness of microtubule bundles at the intercellular bridge were used as criteria to stage telophase cells. Insets show a 3× magnification of the midbody. Scale bars, 10 μm. **b** Time course analysis of protein expression and phosphorylation during mitotic exit after MYPT1 depletion. HeLa cells were treated with siRNAs directed against either a random sequence (control) or *MYPT1* and after 24 h synchronized by thymidine/nocodazole block. Cells were collected at the indicate time points after nocodazole (noc) release and proteins extracted and used in western blot analysis to identify the proteins and phospho-epitopes indicated to the right. The numbers on the left indicate the sizes of the molecular mass marker. Source data for Fig. 5b are provided as a Source Data file

substrates. In line with its established role, our data indicate that MYPT1-PP1β dephosphorylates MRLC to control the contractility of the actomyosin ring during furrow ingression (Fig. 5), but also unexpectedly reveal that MYPT1-PP1β controls the dynamics of central spindle microtubules in late cytokinesis (Fig. 6). Our results suggest that the latter could be mediated, at least in part, through dephosphorylation of MKLP1, and most likely of other MAPs, in order to antagonize Aurora B and possibly other mitotic kinases, like Plk1 and CIT-K, in late cytokinesis (Figs. 6 and 7). We surmise that MKLP1

dephosphorylation by PP1β modulates centralspindlin clustering in order to promote different functions of this complex in late cytokinesis, like its close association with other midbody proteins such as PRC1. This, in combination with dephosphorylation of additional midbody components, would contribute to MYPT1-PP1β-mediated regulation of central spindle microtubule dynamics and midbody architecture in late cytokinesis. In sum, our findings indicate that temporal changes in the spatial distribution of kinases and counteracting phosphatases during cytokinesis control the phosphorylation status, and consequently

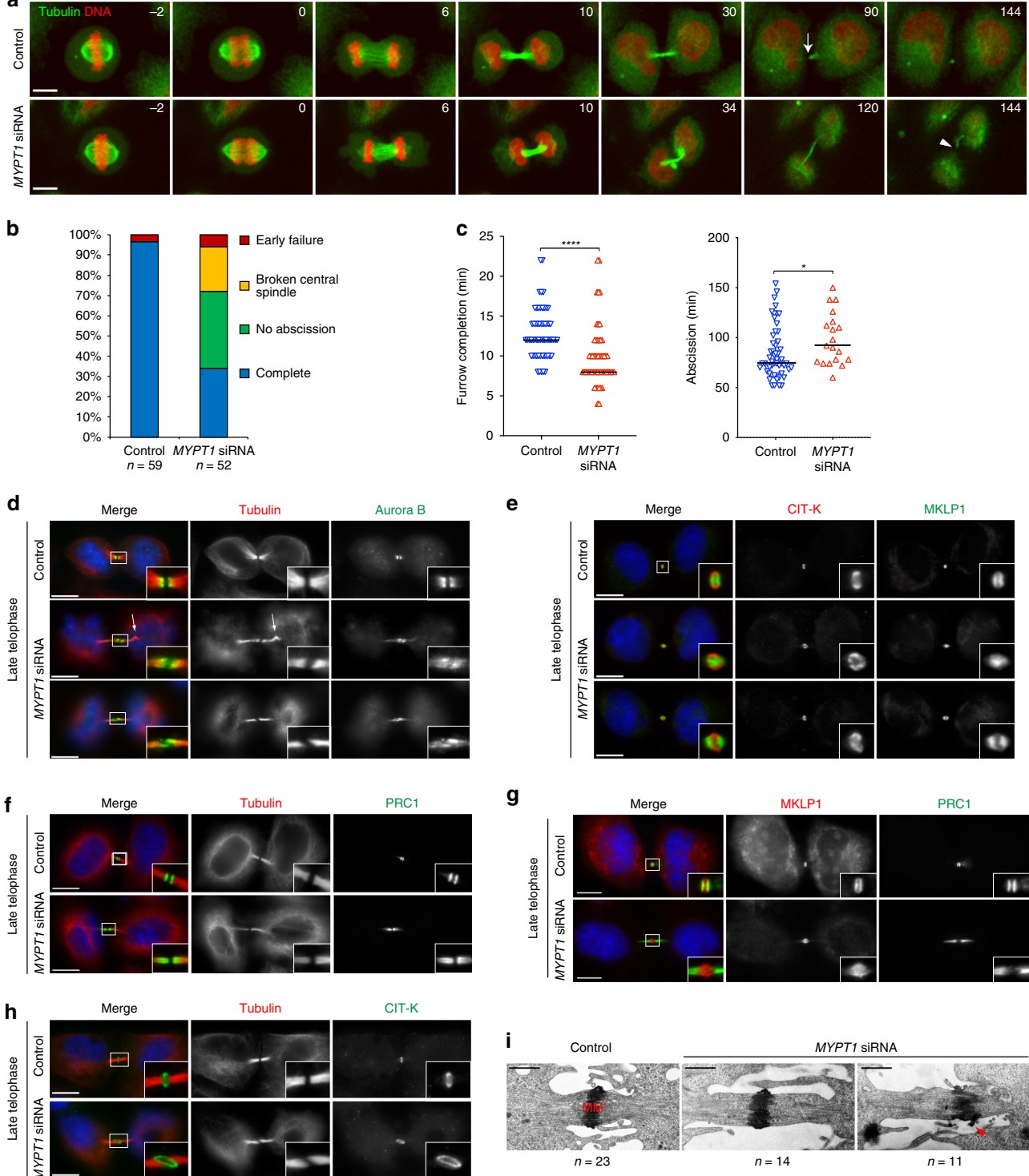

the activity, of cytokinesis proteins as illustrated in Fig. 8. In early telophase MYPT1-PP1β localizes to the cortex of the ingressing furrow (see Fig. 4) where it can de-phosphorylate MRLC to antagonize Rho kinase, but has no or little access to the centralspindlin pool that accumulate at the central spindle midzone, which is instead highly phosphorylated by Aurora B and can therefore form clusters. However, after completion of furrow ingression, MYPT1-PP1β accumulates at the midbody ring whereas Aurora B is slowly degraded and accumulates at the midbody arms. This allows dephosphorylation of MKLP1 at S708, which could strengthen the association of centralspindlin with

other midbody proteins, in particular PRC1. This change in the phosphorylation status of MKLP1, and most likely of other midbody components, is important for maintaining a robust central spindle and for establishing proper midbody architecture.

Finally, our data show that MYPT1 is required for the association of MKLP1 and PP1β in telophase cells (Fig. 7c) and that PP1β requires the VQF sub-optimal binding motif of MKLP1 to select its target S708 residue (Fig. 7). However, our current knowledge of PP1 function indicates that PP1β cannot simultaneously interact with both MYPT1 and MKLP1 through their (R) VxF motifs. One possible explanation could be that MYPT1 is

**Fig. 6** MYPT1 is required for central spindle stability and midbody architecture. **a** Images from time-lapse recordings of HeLa Kyoto cells expressing GFP:: tubulin and H2B::mCherry treated with control siRNAs or *MYPT1* siRNA for 30 h before filming. Time is in min relative to anaphase onset. The arrow in the 90 min control cell marks the abscission site, while the arrowhead in the 144 min *MYPT1* siRNA cell marks the rupture of the central spindle. Scale bar, 10 μm. **b** Graph showing the frequency of phenotypes observed in the time-lapse recordings described in **a**. Categories: no abscission indicates cells that either failed abscission or failed to fully separate during filming (Supplementary Movie 7); early failure indicates cells that failed to form a midbody and cleavage furrows collapsed (Supplementary Movie 8); broken central spindles indicates cells in which the central spindle broke before abscission occurred, like in the cell shown in **a** and in Supplementary Movie 6; $n = 59$ independent control cells and $n = 52$ *MYPT1* siRNA independent cells were counted. **c** Scatter plots showing quantification of furrow ingression (from anaphase onset to furrow completion); $n = 57$ independent control cells and $n = 46$ *MYPT1* siRNA independent cells were counted. Abscission (from furrow completion to abscission) times measured in the time-lapse recordings described in **a**. $n = 57$ independent control cells and $n = 19$ *MYPT1* siRNA independent cells that successfully completed abscission were counted. Horizontal bars indicate medians; ****$p < 0.0001$ (student's T-test); *$p < 0.05$ (Mann–Whitney *U* test). **d–h** HeLa Kyoto cells (control) or *MYPT1* siRNA were stained to detect the indicated epitopes and DNA. DNA condensation and the size of microtubule bundles at the intercellular bridge were used to stage telophase cells. Insets show a 3× magnification of the midbody. The arrow in **d** marks a bend in the central spindle. Scale bars, 10 μm. **i** Electron micrographs of midbodies in HeLa cells control or *MYPT1* siRNA for 48 h. $n = 23$ independent control cells and $n = 25$ *MYPT1* siRNA independent cells. The arrowhead marks an abnormal protrusion of the midbody matrix (MM). Scale bars, 1 μm. Source data for Fig. 6b, c are provided as a Source Data file

necessary to initially bring PP1β in proximity of MKLP1, but then a pool of PP1β could dissociate from MYPT1 to interact with MKLP1 through its less efficient VQF site to de-phosphorylate S708. Future studies can clarify whether such a two-step mechanism of action does indeed exist and how widely it is employed by PP1 catalytic subunits.

## Methods

**Molecular biology.** The coding sequence for PP1β was amplified by PCR using the Addgene plasmid 31677 as a template to create an entry clone in pDONR221 using Gateway technology according to manufacturer's instruction (ThermoFisher). The plasmid pEGFP-C1::MKLP1 was previously described[29]. The QuikChange Lightning Site-Directed Mutagenesis Kit (Agilent) was used to generate the PP1β phosphatase dead mutant (harboring the mutations D94N and H124N) and the pEGFP-C1::MKLP1$^{AQA}$ plasmid containing two substitutions in the VQF$_{786–788}$ binding site for PP1β. The sequence of all DNA constructs were verified by sequencing (Source BioScience). All oligonucleotides used in this study are listed in Supplementary Table 3.

**Cell culture and treatments.** HeLa Kyoto (kind gift from Ina Poser, Max Planck Institute of Molecular Cell Biology and Genetics) were maintained in DMEM (Sigma) containing 10% fetal bovine serum (Sigma) and 1% pennicillin/streptomycin (Invitrogen) at 37 °C and 5% CO$_2$. HeLa cell lines stably expressing GFP or Flag-tagged transgenes (listed in Table S2) were cultured in the same medium with the addition of appropriate selection antibiotics (puromycin and/or G418). The HeLa Kyoto cell line stably expressing GFP::tubulin and H2B::mCherry was described[39]. hTERT RPE-1 cells (ATCC) were cultured in DMEM/F12 (Sigma) containing 2 mM L-glutamine, 10% fetal bovine serum (Sigma), and 1% pennicillin/streptomycin (Invitrogen) at 37 °C and 5% CO$_2$. HeLa S3 (ATCC) were culture in DMEM (Sigma) containing 2 mM L-glutamine, 10% fetal bovine serum (Sigma), and 1% pennicillin/streptomycin (Invitrogen) at 37 °C and 5% CO$_2$.

For RNA interference the following siRNAs were used: scrambled sequence control: 5′-AACGTACGCGGAATACTTCGA-3′, CIT-K: 5′-ATGGAAGGCACTA TTTCTCAA-3′, PP1α: 5′-AAGAGACGCTACAACATCAAA-3′, PP1β: 5′-ACGA GGAUGUCGUCCAGGAA-3′ and 5′-GUUCGAGGCUUAUGUAUCA-3′, PP1γ: 5′-ACAUCGACAGCAUUAUCCAA-3′ and 5′-AGAGGCAGUUGGUCACUCU-3′, MYPT1: 5′-AGUACUCAACCAUAAUUAA-3′, MKLP1 3′UTR: 5′-AAGCAG UCUUCCAGGUCAUCUUU-3′, using Lipofectamine RNAiMAX (Invitrogen) following the manufacturer's instructions. All these siRNAs have been previously validated for specificity and efficacy[10,24,29,40].

Cell lines stably expressing MKLP1 or MKLP1$^{AQA}$ constructs were generated by transfecting $2 \times 10^6$ HeLa Kyoto cells with 10 μg of plasmid DNA using the Neon Transfection System (ThermoFisher) using manufacturer's instructions. After 48 h, cells in 100 mm culture dish were selected in complete selective medium containing 400 μg ml$^{-1}$ G418 for ~2 weeks until colonies became visible. Individual colonies were picked, cultured under resistance and tested for expression of the construct by western blot and immunofluorescence. To generate the cell line expressing Aurora B::GFP, HeLa cells (originally from ATCC) were transfected with the mammalian expression vector pcDNA3.1 containing the coding sequence of human Aurora B, C-terminally fused to GFP, using FuGENE transfection reagent according to the manufacturer's instructions. Twenty-four hours posttransfection G418 was added to the medium (500 μg ml$^{-1}$), and cells incubated for a further 7 days. The population was then expanded and FACS sorted on the GFP signal, and maintained under G418 selection.

HeLa cells were synchronized in S phase by double thymidine block. Cells were first arrested in S phase by the addition of 2 mM thymidine (Sigma-

Aldrich) for 19 h, washed twice with phosphate-buffered saline (PBS) and released for 5 h in fresh complete medium. After release, cells were incubated again for 19 h in complete medium containing 2 mM thymidine, washed twice with PBS, released in fresh medium for 10 min, harvested by centrifugation at $1000 \times g$ for 3 min, washed in PBS, frozen immediately in dry ice and stored at −80 °C. To synchronize HeLa cells in metaphase and telophase, we used a thymidine-nocodazole block and release procedure. Cells were first arrested in S phase by a single thymidine treatment as described above, washed twice with phosphate-buffered saline (PBS) and released for 5 h in fresh complete medium. Cells were then cultured for additional 13 h in fresh complete medium containing 50 ng ml$^{-1}$ nocodazole (Sigma-Aldrich) and then harvested by mitotic shake-off. Mitotic cells were washed three times with PBS, and either released for 30 min in fresh medium containing 10 μM MG132 (Sigma) to collect cells in metaphase or released in just fresh medium for 90 min to collect cells in telophase. Cells were then harvested by centrifugation and frozen in dry ice.

**Midbody purification.** For SILAC experiments, HeLa S3 cells were grown in DMEM lacking Arg and Lys (Invitrogen), and supplemented with 10% (v/v) 1 kDa dialyzed FBS (Sigma-Aldrich), 1% (v/v) penicillin/streptomycin and either unlabeled Arg and Lys or L-[$^{13}C_6$,$^{15}N_4$] Arg and L-[$^{13}C_6$,$^{15}N_2$] Lys (Cambridge Isotope Laboratories) at concentrations of 42 μg ml$^{-1}$ (Arg) and 72 μg ml$^{-1}$ (Lys). Trypsin-EDTA was used to split cells as usual. However, as this solution might contain some non-isotopically labeled amino acids, detached cells were pelleted at $250 \times g$ for 3 min and washed once with sterile phosphate-buffered saline (PBS) before being re-seeded in fresh medium.

To purify midbodies, $2.8 \times 10^7$ HeLa S3 cells were plated into three three-layer tissue culture flasks, 525 cm$^2$ (BD Biosciences), for a total of $8.4 \times 10^7$ cells per condition. Cells were synchronized using the thymidine-nocodazole block and release procedure described in the previous section. After nocodazole washout cells were incubated for 2 h in fresh medium containing 10 μM MG132 (Sigma-Aldrich) to further increase the effectiveness of the synchronization, and then incubated at 37 °C for 80 min after release from MG132. Just before collection, 5 μg ml$^{-1}$ taxol (Sigma-Aldrich) was added to the medium for 2–3 min to stabilize microtubules in vivo. Cells were then transferred into a 50 ml conical tube and collected by centrifugation at $250 \times g$ for 3 min. After one wash with pre-warmed H$_2$O, cells were gently resuspended in 25 ml of swelling solution (1 mM PIPES pH 7.0, 1 mM MgCl$_2$, 5 μg ml$^{-1}$ taxol and Roche Complete Protease Inhibitors) and immediately centrifuged at $250 \times g$ for 3 min. The cell pellet was then resuspended in 40 ml of lysis buffer (1 mM PIPES pH 7, 1 mM EGTA, 1% [v/v] NP-40, 5 μg ml$^{-1}$ taxol, 3 U ml$^{-1}$ DNAse I, 10 μg ml$^{-1}$ RNAse A, 1 U ml$^{-1}$ micrococcal nuclease, and Roche Complete Protease Inhibitors) and vortexed vigorously for 1 min. After the addition of 0.3 volumes of cold 50 mM 2-(N-mopholino)ethanesulfonic acid (MES) pH 6.3, the sample was incubated on ice for 20 min and then centrifuged at $200 \times g$ for 10 min at 4 °C. The supernatant was transferred to a new tube and centrifuged at $650 \times g$ for 20 min at 4 °C to pellet midbodies. The midbody pellet was then resuspended in 4 ml of 50 mM MES pH 6.3 and centrifuged through a 25 ml glycerol cushion (40% [w/v] glycerol diluted in 50 mM MES pH 6.3) at $2800 \times g$ for 45 min at 4 °C. After removal of the glycerol cushion, the midbody pellet was washed with 2 ml of 50 mM MES pH 6.3, transferred to a 15 ml conical tube and centrifuged at $2800 \times g$ for 20 min at 4 °C. For mass spectrometry (MS) analyses, after removing as much liquid as possible, the midbody pellet was resuspended in 100 μl of 50 mM MES pH 6.3 and 900 μl of cold acetone were added to the tube, which was then vortexed and incubated for 10–15 min at −20 °C. The sample was then centrifuged at $3500 \times g$ for 10 min at 4 °C, the supernatant was carefully discarded and the pellet was left to dry for 5–10 min at room temperature (RT). Precipitated proteins were stored at −80 °C until further processing.

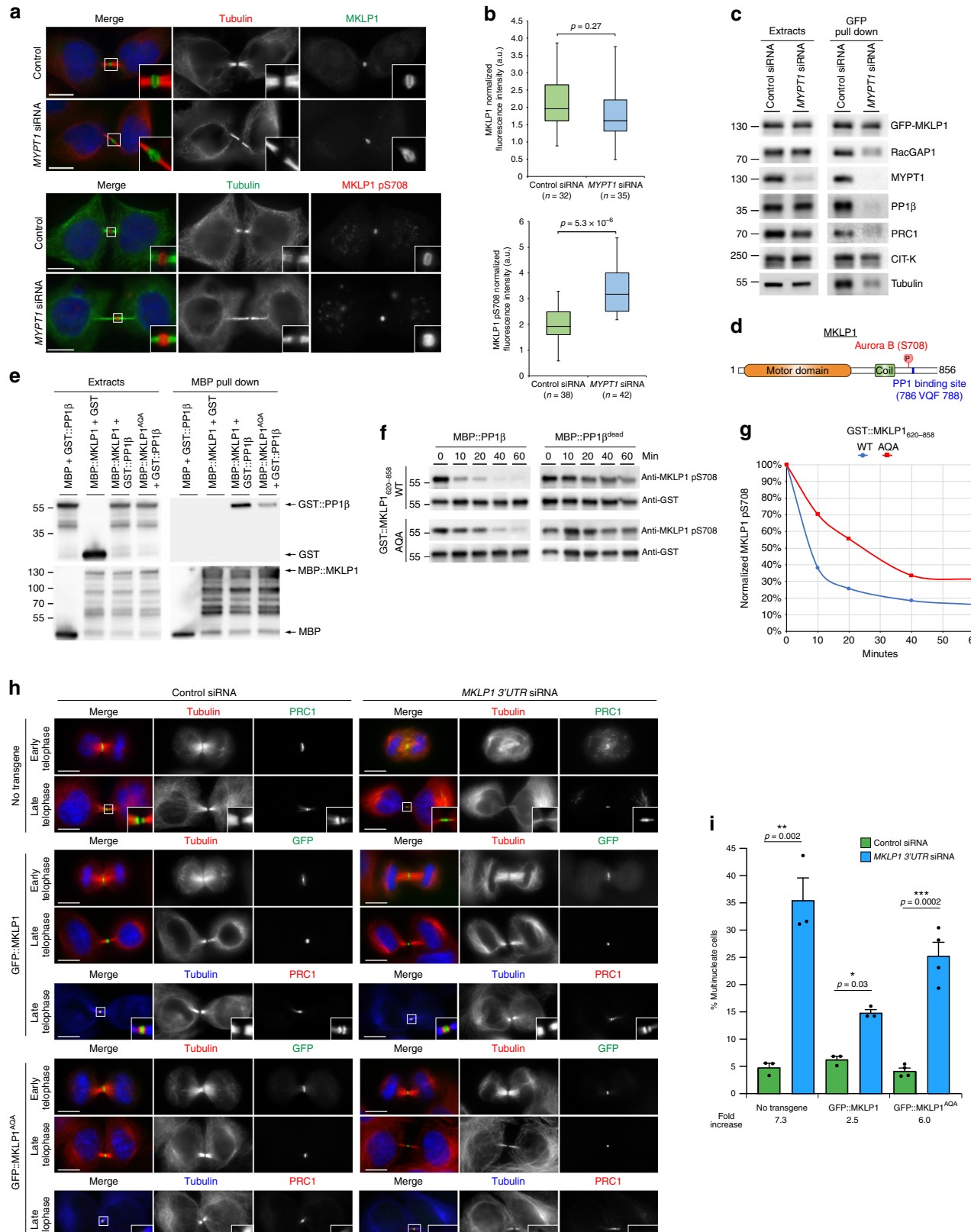

**Affinity purification (AP).** For large-scale AP of GFP-tagged proteins and associated partners, cells were plated at 1/6 confluence in either six 175 cm² flasks or in two three-layer 525 cm² tissue culture flasks (BD Biosciences) and after 24 h synchronized at different stages of the cell cycle as described in the previous section, collected, washed in PBS and cell pellets stored at −80 °C. Each cell pellet was resuspended in 5 ml of lysis buffer (20 mM Tris-HCl, 150 mM NaCl, 2 mM MgCl₂, 1 mM EGTA, 0.1% [v/v] NP-40, 1 mM DTT, 5% [v/v] glycerol and Roche

Complete Protease Inhibitors) and homogenized using a high-performance disperser (Fisher). The homogenate was clarified by centrifugation at 750 × g for 15 min at 4 °C and the supernatant was incubated with 200 µl of GFP-Trap_MA magnetic beads (ChromoTek) for 4 h on a rotating wheel at 4 °C. Beads were then washed four times using a magnetic stand in 10 ml of lysis buffer for 5 min on a rotating wheel at 4 °C, transferred to a new tube and washed one more time in 10 ml of PBS. After removing as much liquid as possible, beads were stored at

**Fig. 7** PP1β dephosphorylates MKLP1 at S708 in cytokinesis. **a** HeLa Kyoto cells were treated with control or *MYPT1*siRNA for 48 h and stained to detect the indicated epitopes. Cells were staged as in Fig. 4. Insets indicate 3× magnification of the midbody. Scale bars, 10 μm. **b** Quantification of total and pS708 MKLP1 in control and *MYPT1*siRNA cells. The boxes indicate the first quartile to the third quartile, the horizontal lines the median and the whiskers the minimum or maximum. AU, arbitrary unit; $n = 32$ independent control cells and $n = 35$ *MYPT1* siRNA independent cells for MKLP1 stained cells; $n = 38$ independent control cells and $n = 42$ *MYPT1* siRNA independent cells for MKLP1-pS708 stained cells; $p$ values from student's T-test. **c** HeLa stably expressing GFP-MKLP1 were treated with control or *MYPT1* siRNA, synchronized in telophase and GFP pull-down protein extracts analyzed by western blot. The numbers indicate the sizes of the molecular mass marker. **d** Schematic diagram of MKLP1 protein. The Aurora B phosphorylation site and the VQF PP1-binding site are indicated. **e** MBP-tagged MKLP1, MKLP1$^{AQA}$ or MBP alone were co-expressed in yeast and used for MBP pull-down assay. Extracts and pull downs were analyzed by western blot to detect GST and MBP. Numbers indicate the size of the protein ladder. **f** In vitro phosphatase assay of GST-tagged WT and AQA MKLP1. The reactions were incubated with either MBP-tagged PP1β or a catalytically dead version for the times indicated at the top and analyzed by western blot using antibodies against MKLP1 pS708 and GST. **g** Graph showing the normalization of MKLP1 pS708 values against the amounts of GST-MKLP1$_{620-858}$. **h** HeLa Kyoto cells stably expressing GFP-MKLP1, GFP-MKLP1$^{AQA}$ or no transgene were treated with either control or *MKLP1* 3′UTR siRNA were stained to detect the indicated epitopes. Scale bars, 10 μm. **i** Quantification of multinucleate cells from the experiments shown in **h**. More than 500 independent cells were counted in $n \geq 3$ independent experiments. Bars indicate standard errors. \*$p < 0.05$, \*\*$p < 0.01$, \*\*\*$p < 0.001$ (Mann–Whitney *U* test). Source data for Fig. 1b-c, e–g and i are provided as a Source Data file

---

−80 °C before being analyzed by liquid chromatography coupled with tandem MS (LC-MS/MS; see section below). Most of the AP experiments were carried out in duplicates, with the exception of CIT-K::AcGFP, which was in triplicate for each cell cycle phase, and GFP::PRC1, which was performed only once.

**Mass spectrometry (MS) analyses**. For the analysis of AP samples, beads were digested with trypsin and processed as previously described[10]. To analyze SILAC midbody samples, proteins were resuspended in lysis buffer (100 mM Tris pH 8.5, 100 mM tris[2-carboxyethyl]phosphine, 4% SDS [w/v], 8 M urea) and alkylated by the addition of iodoacetamide at a final concentration of 40 mM for 30 min at room temperature in the dark. Samples were then mixed with NuPAGE LDS 4× Sample Buffer (Invitrogen), boiled for 5 min at 90 °C and loaded on NuPAGE Novex 4–12% Bis-Tris Protein Gels (Invitrogen). Gels were then fixed for 30 min at RT in fixing solution (40% [v/v] methanol, 2% [v/v] acetic acid) and stained overnight with Brilliant Blue G-Colloidal Concentrate (Sigma-Aldrich), according to the manufacturer's instructions. Each gel lane was then excised into ~10 bands. Gel bands were cut into smaller pieces, de-stained completely in 50 mM ammonium bicarbonate/50% (v/v) acetonitrile at 37 °C, and dehydrated in pure acetonitrile for 15 min. Gel pieces were then rehydrated in 50 mM ammonium bicarbonate and digested with Trypsin (Sequencing Grade, Roche) overnight at 37 °C. Peptides were extracted from gel pieces twice for 30 min at 37 °C in 50% (v/v) acetonitrile/0.5% (v/v) formic acid, dried in a SpeedVac (Thermo Scientific), and resuspended in 0.5% (v/v) formic acid. For LC-MS/MS analysis of SILAC samples, an LTQ Orbitrap Velos mass spectrometer (Thermo Scientific) coupled with an Ultimate 3000 Rapid Separation LC nano ultra high pressure HPLC system (Dionex) was used. Peptides were loaded and desalted on a PepMap C18 precolumn (5μ-beads, 100 μm × 20 mm, Dionex) and then separated on a PepMap analytical column (2 μm beads, 75 μm × 50 cm, Dionex) over a 60 min linear gradient (90 min/cycle) of 4–34% (v/v) acetonitrile/0.1% (v/v) formic acid at a flow rate of 0.3 μl min$^{−1}$. The LTQ Orbitrap Velos mass spectrometer was operated in the top 10 data-dependent acquisition mode where the preview mode was disabled. The Orbitrap full scan was set at *m/z* 380–1600 with a resolution of 60,000 at *m/z* 400. The 10 most abundant multiply charged precursor ions, with a minimal signal above 2000 counts, were dynamically selected for collision-induced dissociation fragmentation (MS/MS) in the LTQ Velos ion trap, and the dynamic exclusion was set ± 20 ppm within 45 s. The AGC and maximum injection time for Orbitrap were set at 1e6 and 200 ms, and 1e4 and 150 ms for ion trap.

**MS data analysis**. For protein identification in the SILAC experiments, the raw files were processed using MaxQuant (version 1.4.0.8) and the Andromeda search engine[41–43]. In all cases, peptides were searched against the UniProt human database concatenated with reversed copies of all sequences and supplemented with frequently observed contaminants. The following search parameters were used: full trypsin specificity was required, a maximum of two missed cleavages were allowed, carbamidomethyl (Cys) was set as a fixed modification, whereas acetylation (Protein N-term), oxidation (Met), deamidation (Asn/Gln), and carbamylation (Lys and N-terminus of protein) were considered as variable modifications. Maximum protein and peptide false discovery rates (FDRs) were set to 1% and minimum required peptide length was set to seven amino acids. Quantification of proteins in the SILAC experiment was performed using MaxQuant[41]. SILAC multiplicity was set to doublets where Lys0/Arg0 and Lys8/Arg10 were selected as light and heavy labels, respectively. Peptides considered for quantification were unique and razor peptides including unmodified, and modified with carbamidomethylated (Cys), acetylated (Protein N-term), oxidated (Met), carbamylated (Lys and N-term), and deamidated (Asn/Gln). The re-quantification feature was enabled. Statistical evaluation of MS results generated by MaxQuant was performed using Perseus[41].

For the identification of proteins from AP experiments, raw MS/MS data were analyzed using the MASCOT search engine (Matrix Science). Peptides were searched against the UniProt human sequence database and the following search parameters were employed: enzyme specificity was set to trypsin, a maximum of two missed cleavages were allowed, carbamidomethylation (Cys) was set as a fixed modification, whereas oxidation (Met), phosphorylation (Ser, Thr and Tyr), and ubiquitylation (Lys) were considered as variable modifications. Peptide and MS/MS tolerances were set to 25 parts per million (ppm) and 0.8 daltons (Da). Peptides with MASCOT Score exceeding the threshold value corresponding to < 5% false positive rate, calculated by MASCOT procedure, and with the MASCOT score above 30 were considered to be positive.

**Computational and statistical analyses**. We used in-house written Perl scripts to combine the Mascot data from the replicates of AP-MS experiments for each bait and to compare them with datasets obtained from AP-MS experiments using HeLa cells expressing GFP alone at the same cell cycle stage (S phase, metaphase, or telophase) in order to eliminate non-specific hits. Prey hits absent from these GFP negative controls were classed as being specific. Additional common contaminants, such as keratins and hemoglobin, were eliminated manually. The filtered data (Supplementary Data 3) were then analyzed and visualized using Cytoscape (version 3.7.0).

To generate the serine/threonine phosphorylation sub-network, we searched the interactome dataset for proteins whose Uniprot protein names field contained the terms kinase and phosphatase but not tyrosine via grep in the Unix command line. This generated a dataset of 190 proteins that was subsequently manually curated to eliminate proteins that were not directly involved in phosphorylation/dephosphorylation, such as kinase-associated proteins for example. The final list of 136 proteins was entered into a raw tab-delimited text file and then imported into Cytoscape to generate the network shown in Fig. 3d.

GO enrichment analysis was performed using PANTHER[44]. Prism8 (GraphPad) and Excel (Microsoft) were used for statistical analyses and to prepare graphs.

**Time-lapse imaging**. For time-lapse experiments, HeLa Kyoto cells expressing GFP:tubulin and H2B::mCherry were plated on an open μ-Slide with 8 wells (Ibidi, 80826) 30 h after RNAi treatment. Imaging was performed on a Leica DMi8 CS AFC Motorised Research Inverted Digital Microscope. Images were collected with a 40 × /1.30 NA HC Plan APO CS2 - OIL DIC 240 μm objective and excitation Lasers of Argon (65 mW, 488 nm) and of DPSS (20 mW, 561 nm). We used the Application Suite X software (LAS-X; Leica) for multidimensional image acquisition. Specimens were maintained at 37 °C and 5% CO$_2$ via a chamber, and z-series of 14, 1-μm sections were captured at 2 min intervals. All images were processed using Fiji[45] to generate maximum intensity projections, to adjust for brightness and contrast, and to create the final movies.

**Fluorescence microscopy**. HeLa cells were grown on microscope glass coverslips (Menzel-Gläser) and fixed in either PHEM buffer (60 mM Pipes, 25 mM Hepes pH 7, 10 mM EGTA, 4 mM MgCl$_2$, 3.7% [v/v] formaldehyde) for 12 min at room temperature or in ice-cold methanol for 10 min at −20 °C. They were then washed three times for 10 min with PBS and incubated in blocking buffer (PBS, 0.5% [v/v] Triton X-100 and 5% [w/v] BSA) for 1 h at room temperature. Coverslips were incubated overnight at 4 °C with the primary antibodies indicated in the figure legends, diluted in PBT (PBS, 0.1% [v/v] Triton X-100 and 1% [w/v] BSA). The day after, coverslips were washed twice for 5 min in PBT, incubated with secondary antibodies diluted in PBT for 2 h at RT and then washed twice with PBT and once with PBS. Coverslips were mounted on SuperFrost Microscope Slides (VWR) using VECTASHIELD Mounting Medium containing DAPI (Vector Laboratories). Phenotypes were blindly scored by at least two people independently. Images were

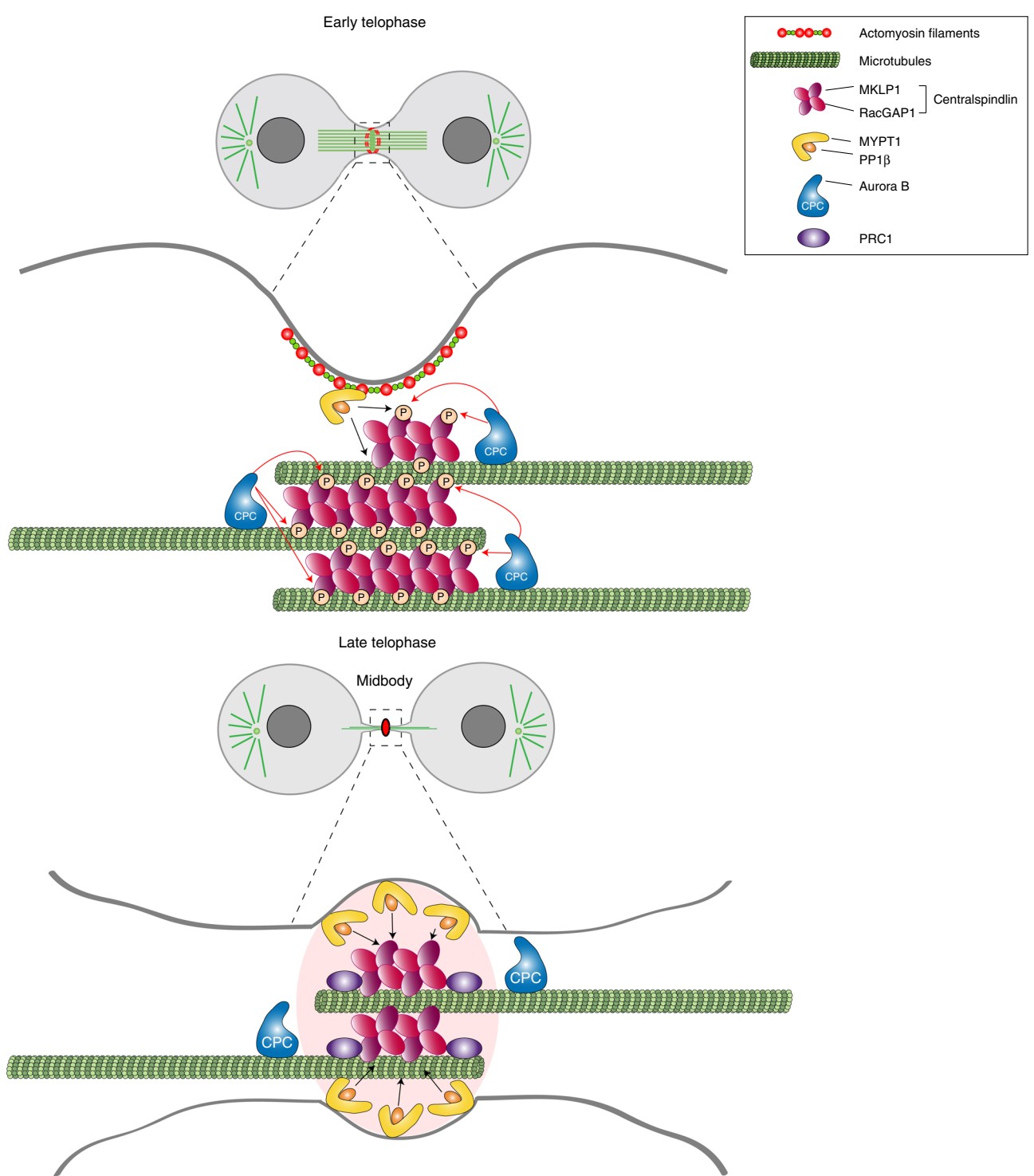

**Fig. 8** Model of regulation of centralspindlin by Aurora B and MYPT1-PP1β during cytokinesis. During furrowing (top panel) MYPT1-PP1β accumulates at the cortex with no or very limited access to the centralspindlin pool that localizes to the central spindle midzone, which is highly phosphorylated by Aurora B and therefore can form clusters. After completion of furrow ingression, MYPT1/PP1β accumulates at the midbody ring whereas Aurora B localizes to the midbody arms (bottom panel). This allows PP1β to de-phosphorylate MKLP1 at S708, which could strengthen the association of centralspindlin with PRC1. See text for more details

acquired using a Zeiss Axiovert epifluorescence microscope equipped with Meta-Morph software. Fiji[45] was used to generate maximum intensity projections, which were adjusted for contrast and brightness and assembled using Photoshop. Fluorescence intensity values in Fig. 7b were measured from identically sized areas at the midbody ($I_M$), in the nucleus ($I_N$), and in the background ($I_B$) using Fiji[45] and then normalized values were calculated using the following formula: $[(I_M-I_B)-(I_N-I_B)]/(I_N-I_B) = (I_M-I_N)/(I_N-I_B)$.

**Antibodies**. The following antibodies and dilutions for western blot (WB) and immunofluorescence (IF) were used in this study: mouse monoclonal anti α-tubulin (clone DM1A, Sigma, T9026 dilutions for WB 1:20,000, for IF 1:2000), rabbit polyclonal anti-β-tubulin (Abcam, ab6046 dilutions for WB 1:5000, for IF 1:400), mouse monoclonal anti-cyclin B1 (clone GNS1, Santa Cruz, sc-245 dilution for WB 1:2000), mouse monoclonal anti-PP1α (clone G-4, Santa Cruz, sc-271762 dilutions for WB 1:1000, for IF 1:50), mouse monoclonal anti-PP1β (clone A-6,

Santa Cruz, sc-365678 dilutions for WB 1:10,000, for IF 1:50), mouse monoclonal anti-PP1γ (clone A-4, Santa Cruz, sc-515943 dilutions for WB 1:2000, for IF 1:50), mouse monoclonal anti-CIT-K (BD Transduction Laboratories, 611377 dilutions for WB 1:1500, for IF 1:250), rabbit polyclonal anti-MYPT1 (clone C-6, Santa Cruz, sc-514261 dilutions for WB 1:1500, for IF 1:50), rabbit polyclonal anti-Aurora A (Abcam, ab1287 dilution for WB 1:4000), rabbit polyclonal anti-TPX2 (Novus Biologicals, NB500-179 dilution for WB 1:1000), rabbit polyclonal anti-KIF14 (Bethyl Laboratories A300-233A dilution for WB 1:2000) rabbit polyclonal anti-MKLP1 (clone N19, Santa Cruz Biotechnology, sc-867, dilutions for WB 1:3000, for IF 1:500), rabbit polyclonal anti-phospho MKLP1 pS708[29] (dilutions for WB 1:2000, for IF 1:200), rabbit polyclonal anti-tri-phospho CHMP4C pS210 pS214 pS215[27] (dilution for WB 1:400), mouse monoclonal anti-Aurora B (clone AIM-1, BD Transduction Laboratories, 611082 dilutions for WB 1:2000, for IF 1:100), mouse monoclonal anti-PRC1 (clone C-1, Santa Cruz, sc-376983 dilutions for WB 1:5000, for IF 1:100), rabbit monoclonal anti-phospho PRC1 pT481 (Abcam, ab62366 dilution for WB 1:12,000), rabbit polyclonal anti-phospho-histone H3 pS10 (Merck, 06-570 dilution for WB 1:10,000), rabbit polyclonal anti-mono-phospho MRLC pS19 (Cell Signaling Technology, 3671, dilutions for WB 1:1000, for IF 1:50), rabbit polyclonal anti-di-phospho MRLC pT18 pS19 (Cell Signaling Technology, 3674, dilutions for WB 1:1000, for IF 1:100), goat polyclonal anti-RacGAP1 (Abcam, ab2270 dilution for WB 1:1000), mouse monoclonal anti-GST (Abcam, ab92 dilution for WB 1:20,000), mouse monoclonal anti-MBP (NEB, E8032 dilution for WB 1:10,000). Peroxidase and Alexa-fluor conjugated secondary antibodies were purchased from Jackson Laboratories and ThermoFisher, respectively.

**Transmission electron microscopy.** For electron microscopy analyses, HeLa Kyoto cells were plated on microscope glass coverslips (Menzel-Gläser). Cells were fixed overnight at 4 °C in 2.5% [v/v] glutaraldehyde in PBS, post fixed for 1 h in 1% [v/v] OsO$_4$ in PBS, dehydrated in a graded series of alcohols, embedded in Epon-Araldite resin, and polymerized for 2 days at 60 °C. Glass slides were separated from the resin after a short immersion in liquid nitrogen. Sections were obtained with a LKB ultratome, stained with uranyl acetate and lead citrate, and observed and photographed with a FEI Tecnai G2 Spirit transmission electron microscope operating at an accelerating voltage of 100 kV and equipped with a Morada CCD camera (Olympus).

**Protein expression time course and GFP pull-down assay.** $1.4 \times 10^6$ HeLa Kyoto cells were plated in large—175 cm$^2$—flasks and transfect with siRNAs (800 pmol) directed against either a scrambled sequence (control) or MYPT1. After 24 h, cells were synchronized by thymidine/nocodazole block, released in fresh medium and divided into four 75 cm$^2$ flasks, and collected after 0, 45, 90, and 120 min as indicated in Fig. 7. Proteins were then extracted, separated on a SDS-PAGE gel, transferred onto PVDF membrane, and probed to detect the antigens indicated in Fig. 7. Uncropped versions of all blots can be found in the Source Data file.

For GFP pull-down assay, GFP::MKLP1 HeLa cells were transfected with siRNAs and synchronized in late telophase (120 min after nocodazole release) as described above. Cells were then collected, washed in PBS, frozen in dry ice and stored at −80 °C. Cell pellets were resuspended in 0.5 ml of lysis buffer (20 mM Tris-HCl, 150 mM NaCl, 2 mM MgCl$_2$, 1 mM EGTA, 0.1% [v/v] NP-40, 1 mM DTT, 5% [v/v] glycerol, Roche Complete Protease Inhibitors and PhosSTOP Protein Phosphatase Inhibitors) and homogenized using a high-performance disperser (Fisher). The homogenate was clarified by centrifugation at $750 \times g$ for 15 min at 4 °C and the supernatant was incubated with 40 μl of GFP-Trap MA magnetic Beads (ChromoTek) for 4 h on a rotating wheel at 4 °C. Beads were then washed four times in 1 ml of lysis buffer for 5 min on a rotating wheel at 4 °C, transferred to a new tube and washed one more time in 1 ml of PBS. After removing as much liquid as possible, beads were resuspended in 2× Laemmli sample buffer (Sigma-Aldrich), boiled for 5 min and stored at −20 °C. Proteins were separated by SDS-PAGE, transferred onto PVDF membrane, and probed to detect the antigens indicated in Fig. 7c. Uncropped versions of all blots can be found in the Source Data file.

**Protein purification and binding assays in yeast.** Interaction between MBP-MKLP1 and GST-PP1β was explored in *S. cerevisiae* using a modification of the method described[46]. Briefly, MBP-MKLP1 proteins (wild-type and AQA mutant) were expressed using a derivative of plasmid pMH919 carrying MBP instead of 6His and GST-PP1β using pMH925. Wild type and mutant alleles were amplified by PCR with oligos containing at their 5′ 40 bps homologous to pMH919 and pMH925, respectively. All expressing plasmids were obtained by recombination (GAP repair)[47] using strains MGY140 for the MKLP1 constructs and MGY139 [Mat a, ade5, ura3-5, trp1-289, his3, leu2, lys2Δ0, mob1::kanMX4, cdc28::LEU2, pep4::LYS2/YCplac33-MOB1-CDC28] for the PP1β construct. After recombination, strains were crossed and passed on FOA plates to obtain diploid strains stably expressing pairs of MKLP1 and PP1β proteins (see below). For expression and purification, strains were cultivated for 6 h in YPGal to induce the expression of the recombinant proteins. Cell pellets were resuspended in Breaking Buffer (50 mM Tris-HCl pH 7.5, 250 mM NaCl, 10% glycerol, 0.2% [v/v] NP-40, 5 mM EDTA, 5 mM DTT, Protease Inhibitor cocktail and 1 mM PMFS) and crude extract was obtained by vortexing in presence of glass beads

(0.5 mm diameter). MBP-MKLP1 proteins were purified on amylose resin (New England Biolabs), washed five times with Washing Buffer (50 mM Tris-HCl pH 7.5, 250 mM NaCl, 0.2% [v/v] NP-40, 5 mM DTT) and eluted with 20 mM maltose. The presence of GST-PP1β bound to MKLP1 was revealed by western blotting using an anti-GST antibody. Uncropped versions of all blots can be found in the Source Data file.

PP1β and PP1β$^{dead}$ proteins were expressed and purified as MBP-fusion proteins in *S. cerevisiae* essentially as described above. Briefly, MBP-PP1β proteins were expressed using a derivative of plasmid pMH919 carrying MBP instead of 6His. PP1β and PP1β$^{dead}$ coding sequences were amplified by PCR with oligonucleotides containing at their 5′ 40 bps homologous to pMH919. Expressing plasmids were obtained by recombination using strain MGY70 (see below). For expression and purification, strains were cultivated for 6 h in YPGal to induce the expression of the recombinant proteins. Cell pellets were resuspended in Breaking Buffer and crude extract was obtained by vortexing in presence of glass beads (0.5 mm diameter). MBP-PP1β proteins were purified on amylose resin and eluted with 20 mM maltose as described above. These are the *S. cerevisiae* strains generated during this study: MGY930, to express MBP/GST-PP1β; MGY931 to express MBP-MKLP1 and GST; MGY932 to express MBP-MKLP1 and GST-PP1β; MGY951 to express MBP-MKLP1$^{AQA}$ and GST-PP1β; MGY960 to express MBP-PP1β; and MGY961 to express MBP-PP1β$^{dead}$.

**In vitro GST pull-down and phosphatase assays.** DNA fragments coding for MKLP1$_{620–858}$ (wild type and AQA mutant) were generated by PCR and cloned into pDEST15 (ThermoFisher) to express N-terminal GST-tagged polypeptides in *E. coli*. GST-tagged products were then purified using Glutathione Sepharose 4B according to manufacturer's instruction (GE Healthcare).

For GST pull downs, MBP::PP1β purified from yeast and eluted from beads (see previous section) was mixed with 25 μl of Glutathione Sepharose beads containing purified GST or GST::MKLP1$_{620–858}$ proteins (wild type and AQA mutant). Samples were incubated in 300 μl of NET-N + buffer (50 mM Tris-HCl, pH 7.4, 150 mM NaCl, 5 mM EDTA, 0.2% NP-40, and a cocktail of Roche Complete Protease Inhibitors) for 60 min at 4 °C on a rotating wheel and then washed five times with 500 μl of wash buffer (50 mM Tris-HCl, pH 7.4, 250 mM NaCl, 5 mM EDTA, 0.2% NP-40, and a cocktail of Roche Complete Protease Inhibitors) followed by centrifugation at $500 \times g$ for 1 min. Beads were resuspended in 25 μl of Laemmli SDS-PAGE sample buffer and typically 1/5 was loaded on a 4–20% Tris-Glycine gel for western blot analysis. Uncropped versions of all blots can be found in the Source Data file.

For the phosphatase assay, 10 μg of purified GST-MKLP1$_{620–858}$ proteins (wild-type and AQA mutant) bound to beads were phosphorylated in vitro with 2 μg of recombinant His-tagged Aurora B (ThermoFisher) in 40 μl final volume of kinase buffer (20 mM HEPES pH 7.5, 5 mM MgCl$_2$, 1 mM DTT, 0.1 mM cold ATP) at 30 °C for 30 min and then washed three times with 500 μl of phosphatase buffer (50 mM Tris-HCl pH 7.5, 100 mM NaCl, 1 mM MnCl$_2$) to remove His-tagged Aurora B. Beads were then resuspended in 100 μl of phosphatase buffer, divided in 10 aliquots, and each aliquot was incubated with 200 ng of either MBP-PP1β or MBP-PP1β$^{dead}$ enzymes purified from yeast (see previous section) at 30 °C with gentle agitation. Samples were collected at 10, 20, 40, and 60 min and reactions were immediately stopped with the addition of 2× Laemmli buffer. Proteins were separated by SDS-PAGE and analyzed by western blot with anti-phospho MKLP1 pS708 and anti-GST antibodies. Uncropped versions of all blots can be found in the Source Data file. Chemiluminescent signals were acquired below saturation levels using a G:BOX Chemi XRQ (Syngene) and quantified using Fiji[45].

**Reporting summary.** Further information on research design is available in the Nature Research Reporting Summary linked to this article.

## Data availability

The SILAC mass spectrometry proteomics data have been deposited to the ProteomeXchange Consortium via the PRIDE partner repository with the dataset identifier PXD012374. The other proteomics data supporting the findings of this study are available within the paper and its supplementary information. All other data, materials and reagents are available from the corresponding author upon reasonable request. The source data underlying Figs. 1a, 1e, 4e, 4h, 5b, 6b, c, 7b, c, 7e–g, 7i, and Supplementary Figs. 1, 3a, 3c and 5c are provided as a Source Data file.

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

## Acknowledgements

We are very grateful to D.M. Glover for allowing us free access to his microscopy facility, to P.A. Coelho for help with time-lapse experiments and to Z. Lipinszki for helpful discussions. We thank D. Gerlich, A.A. Hyman, I. Poser, M. Petronczki, and E. Zanin for cell lines. This work was funded by a BBSRC grant (BB/R001227/1) to P.P.D. and work in JC lab is supported by a CR-UK centre grant (C309/A25144).

## Author contributions

Conceptualization: P.P.D., L. Capalbo and M.M.; Investigation: Z.I.B., L. Capalbo, P.P.D., M.G., M.E.D., S.T., L. Copoiu, G.C., M.G.R. and L.Y.; Formal analysis: Z.I.B., L. Capalbo, P.P.D., A.E. and E.F.; Funding acquisition: P.P.D.; Methodology: Z.I.B., L. Capalbo and P.P.D.; Project administration: P.P.D.; Resources: J.C., M.E.D. and M.M.; Supervision: P.P.D.; Validation: Z.I.B., L. Capalbo and P.P.D.; Visualization: Z.I.B., L. Capalbo, A.E. and P.P.D.; Writing—original draft: P.P.D.; Writing—review and editing: L. Capalbo, P.P.D., M.G., J.C., M.M. and S.W.

## Competing interests

The authors declare no competing interests.
