## [Peer Review File · Nature Communications]

Reviewers' comments:

Reviewer #1 (Remarks to the Author):

This manuscript "The midbody interactome reveals new unexpected roles for PP1 phosphatases in cytokinesis" from Capalbo et. al. describes a proteomics study using affinity purification of tagged midbody proteins at various cell cycle stages. The dataset from this study updates an earlier study from 2004 to provide a more comprehensive interactome.

I thought the manuscript was straightforward and easy to understand. The siRNA validation of the phosphatases seemed to adequately validate the finding of the PP1 β /MYPT1 phosphatase's role in cytokinesis and its dephosphorylation of MKLP1 was also a nice finding.

I do have some minor concerns/questions for the authors. I recommend this manuscript for publication in Nature Communications after these issues are addressed.

Questions:

CIT-K-GFP to identify the CIT-K interactomes at different cell cycle stages - S phase, metaphase and telophase by AP-MS. The number of interactors identified in telophase, at 5-10x more than S phase and metaphase, seems strikingly high (or the other two being a little too clean). What evidence do the authors have that this is not artefact? How many interactors would be identified from asynchronous cells, for e.g.?

Why is protein ID done with MaxQuant and Mascot for SILAC and label-free respectively? MaxQuant can do both modes and I would suggest using one search engine for consistency of protein ID/FDR reporting.

Reviewer #2 (Remarks to the Author):

This manuscript deals with the protein interactome of the midbody, as derived from a mass spectrometric analysis of proteins that co-purify with tagged versions of 10 previously established midbody proteins. The data disclose a large and complex network of midbody protein interactions. The authors subsequently focused on the phosphatase regulator MYPT1 and explored its function in the dephosphorylation of the centralspindlin component MLKP1. In general, the data are interesting but not always conclusive. Key controls are missing and some data can be interpreted differently, as detailed below.

1. The authors' claim that their findings expand the temporal window of PP1 during mitosis is not correct as other functions of PP1 in late mitosis (e.g. nuclear envelope re-assembly) and early G1 (e.g. chromatin organization) have already been documented.
2. The reasoning for focusing on MYPT1 is not clear as PP1 β is not in the midbody proteome list (Table S1). PGAM5 and PPP1R9A/B seem equally attractive regulatory midbody components but are not even mentioned in the text.
3. Fig. 2 is not illustrative as it only shows a list of GO terms that cover a diverse number of cellular processes.
4. In the methods section it is stated that 'most affinity purifications were carried out in duplicates'. Does this imply that some experiments were only performed once? In general the legends lack information on the statistics.

5. Fig 1A: which phospho-H3? Fig. 1C: the knockdown of citron kinase clearly affected the distribution of MKLP1 but this is not commented upon.

6. The antibodies used for immunostaining are not properly validated. Are the signals lost after knockdown of the epitope? It is worrisome that the localization of the PP1 isoforms does not agree with published data (e.g association of PP1gamma with chromosomes).

7. The PP1 KD experiments are not conclusive as the PP1 isoforms act redundantly. It therefore cannot be ruled out that PP1alpha and PP1gamma (isoforms of the same subfamily) also have a key function in cytokinesis. Such functions could be disclosed by the combined knockdown of PP1alpha and PP1gamma.

8. In Fig. 5, knockdown experiments with PP1beta are missing: they should give the same result as knockdown of MYPT1. Also, the key effects of the knockdown of MYPT1 and PP1beta on MKLP1 phosphorylation and midbody organization should be rescued by inhibition of Aurora B

9. It cannot be concluded that MKLP1 binds directly to PP1. The conserved 'VQF' motif randomly occurs in a large fraction of proteins and is remote from the consensus RVXF sequence for binding to PP1, which also requires for example an N-terminal basic residue and should reside in a structurally disordered domain. The binding of MKLP1 to PP1 in pull down experiments can be indirect. The reduced binding of mutated MKLP1 to PP1 in pulldown experiments can also be explained by conformational changes. There is also a conceptual problem here: MYPT1 has a RVXF motif that mediates binding to PP1, and prevents the simultaneous binding of a second protein (such as MKLP1) via an RVXF sequence. This makes it very unlikely that MKLP1 is a substrate of the PP1beta-MYPT1 holoenzyme.

10. It cannot be excluded that the effects of PP1-MYPT1 on the phosphorylation of MKLP1 are indirect and, for example, mediated by PP2A, another midbody component (Table S1) and known to be activated by PP1 at the mitotic exit. A mediatory role for PLK1 cannot be excluded either as it is an interactor of MYPT1 and activator of Aurora B.

11. All cell experiments have been performed with HeLa cells. It seems important to explore whether the described function of PP1beta-MYPT1 also applies to (nearly) diploid, non-cancer cells (e.g. RPE1 cells).

12. The mechanism of the different timing of dephosphorylation of MRLC and MKLP1 by PP1-MYPT1 is not elaborated upon.

Reviewer #3 (Remarks to the Author):

This manuscript from the D'Avino group reports proteomic identification of Citron Kinase (CIT-K) interactors performed using affinity purification coupled to mass spectrometry (AP-MS) at different cell cycle stages (S, prometaphase, metaphase, telophase, G1), part of the "midbody interactome". Purified midbodies were also subjected to MS to identify the "midbody proteome". SILAC-based MS was performed on midbodies purified with or without siRNA-mediated depletion of CIT-K, with only minor differences observed between the two conditions. A more complete "midbody interactome" of almost 3000 proteins was then defined by expanding the AP-MS to include 9 additional bait proteins: Anillin, Aurora B, CHMP4B, CHMP4C, Ect2, KIF14, KIF20A/MKLP2, KIF23/MKLP1, PRC1. 1230 proteins were found in both the midbody interactome and the midbody proteome. The authors then generated a "midbody interactome phosphorylation sub-network" by extracting from their dataset the entire interactome of proteins involved in serine/threonine, but not tyrosine, phosphorylation. This led them to PP1 phosphatases of which the three PP1 catalytic subunits - α , β and γ - and the myosin

phosphatase target subunit 1, MYPT1, were all shown to localize to the midbody ring. The rest of the paper then goes on to characterize, in-depth, the role of MYPT1 and the phenotypic consequences to cytokinesis of MYPT1 depletion. Interesting data showing midbody defects upon MYPT1 depletion are presented

Overall, this is a clearly written, high quality manuscript. The data are well presented, appear to be robust and support the main conclusions made. The work represents an important contribution to the field in terms of the identified midbody interactome/proteome components. However, it is the follow-on work that really provides the novel mechanistic insight in the form of the careful analysis of MYPT1 localization and function during unanticipated times and processes during later stages of cytokinesis than those that it is usually associated with.

I recommend publication provided that the following minor points are addressed:

- 1) In Abstract: "...revealed that PP1 β /MYPT1 phosphatase regulates microtubule dynamics in late cytokinesis in part through de-phosphorylation of the kinesin component MKLP1/KIF23 of the centralspindlin complex, a key cytokinesis regulator". "Microtubule dynamics" were not tested and so this claim should be reworded (also bottom of Pg. 9 in the Discussion). The observed differences in tubulin staining in the intercellular bridge might reflect other primary defects.
- 2) What was the cell cycle stage of the AP-MS for the 9 additional baits? Only telophase? This should be made explicitly clear in the results section and Methods.
- 3) How exactly was the "phosphorylation sub-network" extracted from the data? This appears to be lacking and should be briefly but explicitly described in the results with a detailed description in the Methods.
- 4) How conserved is S708? It would be helpful to discuss this and incorporate an alignment of this residue in Fig. S4 alongside the alignment of the VQF motif.
- 5) Pg. 6 "We found that, indeed, that the levels of both". Delete 2nd "that".

Capalbo et al.
Manuscript NCOMMS-19-05597-T
Responses to Reviewers

(Reviewers' comments are in blue and changes in the manuscript are highlighted in yellow)

Reviewer 1

This manuscript "The midbody interactome reveals new unexpected roles for PP1 phosphatases in cytokinesis" from Capalbo et. al. describes a proteomics study using affinity purification of tagged midbody proteins at various cell cycle stages. The dataset from this study updates an earlier study from 2004 to provide a more comprehensive interactome.

I thought the manuscript was straightforward and easy to understand. The siRNA validation of the phosphatases seemed to adequately validate the finding of the PP1 β /MYPT1 phosphatase's role in cytokinesis and its dephosphorylation of MKLP1 was also a nice finding.

I do have some minor concerns/questions for the authors. I recommend this manuscript for publication in Nature Communications after these issues are addressed.

We are pleased that the Reviewer found our study interesting and well done. S/he raised a few questions, which we found to be thoughtful and helpful. Our point-by-point responses to the Reviewer's comments are below.

Questions:

CIT-K-GFP to identify the CIT-K interactomes at different cell cycle stages - S phase, metaphase and telophase by AP-MS. The number of interactors identified in telophase, at 5-10x more than S phase and metaphase, seems strikingly high (or the other two being a little too clean). What evidence do the authors have that this is not artefact? How many interactors would be identified from asynchronous cells, for e.g.?

This is a legitimate concern. However, the AP-MS experiments were carried out in three temporally distinct replicates and in each experiment cells were synchronized at the different cell cycle stages (S phase, metaphase and telophase), collected, and protein extracted and processed for AP-MS in parallel. This approach was designed to reduce variables as much as possible. We consistently identified many more interactors (>600) in telophase than in any of the other two stages. We have now clarified this in the text (page 4). Moreover, to further strengthen our data, we have performed yet another experiment, in which we did not identify any additional hits in S phase and telophase (probably indicating that we have reached saturation), but we found some additional interactors in metaphase, which have been included in both the Supplementary Data S1 file and the Venn diagram shown in the modified Figure 1b. These new results do not affect our overall conclusions.

In addition, we carried out an AP-MS experiment using asynchronous cells as suggested by the Reviewer. This identified a total of 252 proteins, which, as expected, included a mix of proteins identified in the other AP-MS purifications from synchronized cells. We have included an excel file for the Reviewer (Data R1) to look at the data, but we do think that it is worth including them in the manuscript.

We don't know for sure why CIT-K interacts with so many more proteins during telophase, but we have unpublished evidence that the phosphorylation pattern of CIT-K changes from metaphase to telophase, which could well be the reason.

Why is protein ID done with MaxQuant and Mascot for SILAC and label-free respectively? MaxQuant can do both modes and I would suggest using one search engine for consistency of protein ID/FDR reporting.

It is simply for historical reasons. We have carried out AP-MS experiments for more than 15 years and have developed over time computational tools that allow us to quickly and easily combine and compare Mascot data. We do not have the same tools for MaxQuant.

Reviewer 2

This manuscript deals with the protein interactome of the midbody, as derived from a mass spectrometric analysis of proteins that co-purify with tagged versions of 10 previously established midbody proteins. The data disclose a large and complex network of midbody protein interactions.

The authors subsequently focused on the phosphatase regulator MYPT1 and explored its function in the dephosphorylation of the centralspindlin component MLKP1. In general, the data are interesting but not always conclusive. Key controls are missing and some data can be interpreted differently, as detailed below.

We are pleased that the Reviewer found our study interesting. However, s/he made a series of comments about our interpretation of the results and asked for additional controls. Our point-by-point responses to the Reviewer's comments are below.

1. The authors' claim that their findings expand the temporal window of PP1 during mitosis is not correct as other functions of PP1 in late mitosis (e.g. nuclear envelope re-assembly) and early G1 (e.g. chromatin organization) have already been documented.

We believe that, at the time we submitted our manuscript, our statement was fairly accurate. Nuclear envelope re-assembly occurs before completion of furrow ingression and thus before the phenotypes described in Figure 6. We also do not think that chromatin organization can be considered a mitotic event. We were simply making the point that a role for PP1 phosphatases in late cytokinesis had not been described before. However, while our manuscript was under review, a paper was published describing a role for PP1 γ in abscission (Bhowmick et al., 2019, PMID: 30905608, cited in the revised version), and therefore we now simply say that our findings "expand the repertoire of PP1 functions during mitosis" in both the abstract (page 2) and discussion (page 10).

2. The reasoning for focusing on MYPT1 is not clear as PP1 β is not in the midbody proteome list (Table S1). PGAM5 and PPP1R9A/B seem equally attractive regulatory midbody components but are not even mentioned in the text.

The Reviewer is correct in pointing out that our study revealed other interesting phosphatases and indeed we are currently investigating whether PGAM5 and PPP1R9A/B are involved in cell division. We – honestly – presented in Table S1 the full list of serine/threonine phosphatases identified in our study, but we do not feel that it is opportune and appropriate to discuss other potentially interesting phosphatases in this paper.

As for why we focused on MYPT1/PP1 β , we clearly stated that: "top scores include the three PP1 catalytic subunits - α , β and γ - and the PPP1R12A regulatory subunit" (page 6). In addition our choice was also based on the siRNA experiments that showed a strong requirement for both MYPT1 and PP1 β in cytokinesis (Figure 4h). It is also important to clarify that the fact that we failed to identify PP1 β in the midbody proteome does not necessarily imply that this catalytic subunit is not present in the midbody and indeed our localization studies indicate otherwise (Figure 4b).

3. Fig. 2 is not illustrative as it only shows a list of GO terms that cover a diverse number of cellular processes.

We had to use GO-slim terms in Figure 2b because of the large number of proteins identified in our datasets. It would have been virtually impossible to visualize GO enrichment profiles using less broad GO terms. As discussed in the manuscript, the point of this figure is to show that both datasets present similar enrichment profiles and we think that this is quite evident in Figure 2b.

4. In the methods section it is stated that 'most affinity purifications were carried out in duplicates'. Does this imply that some experiments were only performed once? In general the legends lack information on the statistics.

We clearly stated in the Methods that: "Most of the AP experiments were carried out in duplicates, with the exception of CIT-K::AcGFP, which was in triplicate, and GFP::PRC1, which was performed only once." Therefore, the only AP-MS experiment that was performed only once is GFP::PRC1, all the others were in duplicate. We didn't feel the need to repeat the GFP::PRC1 AP-MS because we obtained very good results with almost 900 hits (Supplementary Data S3).

5. Fig 1A: which phospho-H3? Fig. 1C: the knockdown of citron kinase clearly affected the distribution of MKLP1 but this is not commented upon.

We thank the Reviewer for noting this and apologize for the inaccuracy. It is the histone H3 pS10 antibody, which is widely used as mitotic marker. It was already described in the Methods section, but we have now labelled it more clearly also in Figure 1a.

The Reviewer is correct in pointing out that MKLP1 is not properly distributed after CIT-K siRNA. This phenotype has been described and discussed in our two previous publications (Bassi et al., 2013 and McKenzie et al. 2016; both cited in the manuscript) and we do not feel necessary to comment on this again in this paper. However, the comment of the Reviewer made us realize that it would have been helpful to add this information in the manuscript for the readers and thus we have modified the first sentence of the Results: "*Citron kinase (CIT-K) is a contractile ring component that acts as a major midbody organizer by interacting with several midbody components, including the CPC and centralspindlin, and by maintaining their correct localization and orderly arrangement*^{9,10}".

6. The antibodies used for immunostaining are not properly validated. Are the signals lost after knockdown of the epitope? It is worrisome that the localization of the PP1 isoforms does not agree with published data (e.g association of PP1 γ with chromosomes).

We showed in the original version of the manuscript that the signals detected by the four antibodies against the three PP1 catalytic subunit and MYPT1 were strongly reduced after siRNA by Western blot. To further confirm their specificity, we now show that these signals are also reduced at the midbodies in immuno-fluorescence experiments (Figure 4a-d).

We are not sure which publication(s) the Reviewer refers to, but PP1 γ had already been described to localize to the cleavage site (Trinkle-Mulcahy, *et al.*, 2003; now cited in the revised version at page 6) in agreement with our results. We could also detect PP1 γ on chromosomes in anaphase (see Figure R1 below), although not in all cells, as well as in G1 cell nuclei (see late telophase and abscission cells in Figure 4c). Overall, we believe that our results are in good accord with the localization of GFP-PP1 γ described by Trinkle-Mulcahy, *et al.*, (2003) and that minor differences could simply be due to the fixation method that we used (methanol), which is very good in detecting epitopes at the midbody, but not so good in preserving other structures like chromosomes.

7. The PP1 KD experiments are not conclusive as the PP1 isoforms act redundantly. It therefore cannot be ruled out that PP1 α and PP1 γ (isoforms of the same subfamily) also have a key function in cytokinesis. Such functions could be disclosed by the combined knockdown of PP1 α and PP1 γ .

To address the Reviewer's comment, we investigated if the combined knockdown of PP1 α and PP1 γ caused a more significant increase in multinucleation. As suspected by the Reviewer, the combined siRNA knockdown highlighted a potential redundant and/or synergistic role of the two catalytic subunits, albeit the increase in multinucleated cells was still lower than either PP1 β or MYPT1 siRNA. We thank the Reviewer for this suggestion and have included these results in Figure 4h and changed the text in the manuscript accordingly (page 6).

8. In Fig. 5, knockdown experiments with PP1 β are missing: they should give the same result as

knockdown of MYPT1. Also, the key effects of the knockdown of MYPT1 and PP1beta on MKLP1 phosphorylation and midbody organization should be rescued by inhibition of Aurora B

We are puzzled by the first part of this comment because we presented in Figure S3 (Figure S4 in the revised version) that PP1 β depleted cells showed almost identical phenotypes to those observed after MYPT1 siRNA (as shown in Figures 5, 6 and 7a). We now also show an additional image of a PP1 β siRNA cell in Figure 4b to further support our results.

As for the rescue of MYPT1 siRNA phenotypes through Aurora B inhibition, this may sound as an obvious experiment, but in practice it is extremely unlikely to work simply because Aurora B, which is itself necessary for cytokinesis, has numerous functions and substrates and its activity is antagonized by different phosphatases. For example, Aurora B is known to phosphorylate KIF4A, CHMP4C and MKLP1 in cytokinesis. In each case the phosphatase that counteracts Aurora B is different: PP2A-B56 for KIF4A (Bastos et al., 2014, *J Cell Biol* **207**, 683-693), PP1 γ -RIF1 for CHMP4C (Bhowmick et al., 2019, PMID: 30905608) and PP1 β -MYPT1 for MKLP1 (this study). Similarly, PP1 β -MYPT1 has at least two different substrates in cytokinesis, MRLC and MKLP1, and likely more. Thus, it is highly improbable that Aurora B inhibition could rescue the cytokinesis defects caused by MYPT1 or PP1 β siRNA. It is also impossible that Aurora B inhibition could rescue the increase in MKLP1 S708 phosphorylation observed after MYPT1 siRNA simply because Aurora B is necessary to phosphorylate this residue.

In conclusion, it is our opinion that the experiments proposed by the Reviewer - which would require considerable time and effort - could only produce, in a best case scenario, very confusing results.

9. It cannot be concluded that MKLP1 binds directly to PP1. The conserved 'VQF' motif randomly occurs in a large fraction of proteins and is remote from the consensus RVXF sequence for binding to PP1, which also requires for example an N-terminal basic residue and should reside in a structurally disordered domain. The binding of MKLP1 to PP1 in pull down experiments can be indirect. The reduced binding of mutated MKLP1 to PP1 in pulldown experiments can also be explained by conformational changes. There is also a conceptual problem here: MYPT1 has a RVXF motif that mediates binding to PP1, and prevents the simultaneous binding of a second protein (such as MKLP1) via an RVXF sequence. This makes it very unlikely that MKLP1 is a substrate of the PP1beta-MYPT1 holoenzyme.

It is true that our binding assay in yeast (Fig. 7) does not exclude the possibility that an intermediary protein could mediate the interaction between MKLP1 and PP1 β , albeit this does not seem very likely because there is no MKLP1 orthologue in yeast. Therefore, to address the Reviewer's comment we have performed an *in vitro* GST pull down assay using recombinant MKLP1 and PP1 β proteins purified separately from different systems. The results, shown in Figure S5b-c, demonstrate that PP1 β directly binds to the MKLP1 C terminal region that contain the VQF motif and that the binding is reduced when this motif is mutated to AQA.

We explicitly said that the VQF motif is not an optimal PP1 β binding site, but we disagree that is "remote" from the RVxF consensus. We also think that the evidence that this sequence has been conserved during evolution, from worms to humans (Figure S5a), does not support the Reviewer's suggestion that this could be a random occurrence.

It cannot be excluded that changing the VQF sequence could cause a conformational change, but there is no other way to test the role of this motif than mutating it. However, the AQA mutant localizes correctly and can rescue the requirement of MKLP1 in early telophase (Figure 7h), suggesting that, if this mutation did indeed cause a conformational change, this must not be so dramatic to affect the overall structure of MKLP1.

We agree with the Reviewer that PP1 β cannot simultaneously bind to MYPT1 and MKLP1 and indeed we never suggested the existence of such a trimeric complex. Nonetheless, our data show that MYPT1 is required for the association of MKLP1 and PP1 β in telophase cells (Fig. 7c) and that PP1 β requires the VQF sub-optimal binding motif of MKLP1 to select its target S708 residue (Fig. 7). These results would indicate that, at some point, a pool of PP1 β could dissociate from MYPT1

to interact with MKLP1. We did not want to speculate about how and when this could happen because we felt that we did not have sufficient knowledge about the regulation and dynamics of all these factors in late cytokinesis. However, to address the Reviewer's criticism we have included in the revised version a small paragraph at the end of the discussion (page 11) to describe one possible explanation.

10. It cannot be excluded that the effects of PP1-MYPT1 on the phosphorylation of MKLP1 are indirect and, for example, mediated by PP2A, another midbody component (Table S1) and known to be activated by PP1 at the mitotic exit. A mediatory role for PLK1 cannot be excluded either as it is an interactor of MYPT1 and activator of Aurora B.

We show that PP1 β binds directly to MKLP1 and can de-phosphorylate its S708 residue *in vitro* (Fig. 7d-g and S5b-c). These results strongly indicate that PP1 β directly de-phosphorylates MKLP1 and therefore we do not think that speculations about possible indirect effects would be justified.

11. All cell experiments have been performed with HeLa cells. It seems important to explore whether the described function of PP1 β -MYPT1 also applies to (nearly) diploid, non-cancer cells (e.g. RPE1 cells).

We now show in the revised version (new Fig. S3) that siRNA depletion of MYPT1 in RPE-1 cells causes cytokinesis failure and central spindle and midbody defects very similar to those observed in HeLa cells. We believe that these results address the Reviewer's concern.

12. The mechanism of the different timing of dephosphorylation of MRLC and MKLP1 by PP1-MYPT1 is not elaborated upon.

We have included a paragraph in the Discussion (pages 10 and 11) to elaborate on this mechanism.

Reviewer 3

This manuscript from the D'Avino group reports proteomic identification of Citron Kinase (CIT-K) interactors performed using affinity purification coupled to mass spectrometry (AP-MS) at different cell cycle stages (S, prometaphase, metaphase, telophase, G1), part of the "midbody interactome". Purified midbodies were also subjected to MS to identify the "midbody proteome". SILAC-based MS was performed on midbodies purified with or without siRNA-mediated depletion of CIT-K, with only minor differences observed between the two conditions. A more complete "midbody interactome" of almost 3000 proteins was then defined by expanding the AP-MS to include 9 additional bait proteins: Anillin, Aurora B, CHMP4B, CHMP4C, Ect2, KIF14, KIF20A/MKLP2, KIF23/MKLP1, PRC1. 1230 proteins were found in both the midbody interactome and the midbody proteome. The authors then generated a "midbody interactome phosphorylation sub-network" by extracting from their dataset the entire interactome of proteins

involved in serine/threonine, but not tyrosine, phosphorylation. This led them to PP1 phosphatases of which the three PP1 catalytic subunits - α , β and γ - and the myosin phosphatase target subunit 1, MYPT1, were all shown to localize to the midbody ring. The rest of the paper then goes on to characterize, in-depth, the role of MYPT1 and the phenotypic consequences to cytokinesis of MYPT1 depletion. Interesting data showing midbody defects upon MYPT1 depletion are presented

Overall, this is a clearly written, high quality manuscript. The data are well presented, appear to be robust and support the main conclusions made. The work represents an important contribution to the field in terms of the identified midbody interactome/proteome components. However, it is the follow-on work that really provides the novel mechanistic insight in the form of the careful analysis of MYPT1 localization and function during unanticipated times and processes during later stages of cytokinesis than those that it is usually associated with.

I recommend publication provided that the following minor points are addressed:

We are very pleased that the Reviewer found our study convincing and robust and supported its publication. S/he only made some minor comments, which we have addressed in our point-by-point responses below.

1) In Abstract: "...revealed that PP1 β /MYPT1 phosphatase regulates microtubule dynamics in late cytokinesis in part through de-phosphorylation of the kinesin component MKLP1/KIF23 of the centralspindlin complex, a key cytokinesis regulator". "Microtubule dynamics" were not tested and so this claim should be reworded (also bottom of Pg. 9 in the Discussion). The observed differences in tubulin staining in the intercellular bridge might reflect other primary defects.

Our time-lapse experiments (Figure 6a) show that the dynamics of central spindle microtubules is altered in MYPT1 cells. However, the Reviewer is correct in pointing out that we did not demonstrate that de-phosphorylation of MKLP1 is involved in regulating microtubules dynamics. Therefore, we have rephrased two sentences in the Abstract (page 2) and in the Discussion (page 10) as suggested by the Reviewer. We now state in the Abstract that: "*Initial analysis of this interactome already revealed that PP1 β -MYPT1 phosphatase regulates microtubule dynamics in late cytokinesis and de-phosphorylates the kinesin component MKLP1/KIF23 of the centralspindlin complex, a key cytokinesis regulator*", and in the Discussion that; "*Our results suggest that the latter (i.e. central spindle dynamics) could be mediated, at least in part, through de-phosphorylation of MKLP1*".

2) What was the cell cycle stage of the AP-MS for the 9 additional baits? Only telophase? This should be made explicitly clear in the results section and Methods.

Yes, it was only in telophase. We apologize for the missing information and have specified the mitotic stage in the text (page 5).

3) How exactly was the "phosphorylation sub-network" extracted from the data? This appears to be lacking and should be briefly but explicitly described in the results with a detailed description in the Methods.

Again, we are sorry for this oversight. We have included this information in the revised version (pages 6 and 16).

4) How conserved is S708? It would be helpful to discuss this and incorporate an alignment of this residue in Fig. S4 alongside the alignment of the VQF motif.

It is conserved from worms to humans and this was described in Douglas et al., (2010) *Curr Biol*, 20: 927-933. We have now mentioned in the revised text that this residue is evolutionarily conserved (page 8).

5) Pg. 6 "We found that, indeed, that the levels of both". Delete 2nd "that".

We thank the Reviewer for spotting this typo, which has been corrected.

Reviewers' comments:

Reviewer #1 (Remarks to the Author):

The revised manuscript by Capalbo et. al. has addressed my previous concerns. I recommend publication.

Reviewer #2 (Remarks to the Author):

Reviewer 2 is not convinced by the reply to comment 9:

In the new Fig. S5c, an essential control is missing showing that GST-MKLP1 does not bind to the MBP moiety of MBP:PP1b.

PP1 interactors with a functionally validated RVXF motif lose their binding to PP1 (nearly) completely by mutation of the RVXF motif. The PP1:MKLP1 interaction is only partially lost by mutation of this motif (Fig. S5C). This shows that MKLP1 either does not directly bind to PP1 (see previous point) or that MKLP1 has at least one other, more important PP1-binding motif.

All validated PP1-binding RVXF-motifs reside in an intrinsically disordered fragment of the polypeptide. Is the PP1-binding domain of MKLP1 predicted to be disordered? If not, the small reduction in (in)direct PP1:MKLP1 interaction by mutation of the VQF sequence may simply be explained by a conformational change of MKLP1 that partially disrupts its interaction with PP1 or a PP1-binding protein.

Reviewer #3 (Remarks to the Author):

The revised manuscript satisfactorily addresses the points I raised in the first round of review. This is an interesting study and publication is recommended. However, a few additional minor points were noted during this second round of review. For the sake of precision and completeness of the manuscript, it is recommended that the following be addressed, although I do not need to approve these changes:

1) Statistical significance of any observed differences (in frequency of multinucleated cells) from the control should be added to graphs in Figures 4H and 7I.

2) Figure 7I The graph legend (purple) is mislabelled- "MYPT1" instead of "MKLP1".

3) Pg 7 "...the levels of both mono(pS19)- and di(pT18 pS19)-phosphorylated MRLC levels were elevated in MYPT1 depleted cells (Fig 5a-b), which had also an abnormal cytoskeleton and numerous cortical blebs (Fig. 4F)." Legend of Figure 4. "Note that MYPT1 siRNA cells show abnormal cell and nuclear shape and disorganized microtubule and actomyosin cytoskeletal filaments."

The claimed cortical blebs are not evident from Figure 4F, maybe if actin had been stained, and only then could a claim about the actin filaments be made. It is suggested that this be re-worded.

Capalbo et al.
Manuscript NCOMMS-19-05597-T
Responses to Reviewers

(Reviewers' comments are in blue and changes in the manuscript are highlighted in yellow)

Reviewer 1

The revised manuscript by Capalbo et. al. has addressed my previous concerns. I recommend publication.

We are very pleased that the Reviewer is satisfied by the changes we made to the revised manuscript. S/he now recommends publication.

Reviewer 2

Reviewer 2 is not convinced by the reply to comment 9:

In the new Fig. S5c, an essential control is missing showing that GST-MKLP1 does not bind to the MBP moiety of MBP:PP1b.

We strongly disagree that the control requested by the Reviewer is essential because we demonstrated the interaction between MKLP1 and PP1 β in two different systems and using complementary tagging methods: in yeast MBP-MKLP1 interacts with GST-PP1 beta (and the interaction is reduced with the MKLP1 mutant; Fig. 7e) and *in vitro* GST-MKLP1 interacts with MBP-PP1 β (and the interaction is reduced with the MKLP1 mutant; Fig. S5c). The really important control in Fig S5c is that GST alone does not significantly pull down MBP-PP1 β . However, after a specific request from the Editor, we have repeated this experiment to include the pull down of GST-MKLP1 (WT) with the MBP tag alone. As expected, GST-MKLP1 does not interact with the MBP moiety (new Fig. S5c).

PP1 interactors with a functionally validated RVXF motif lose their binding to PP1 (nearly) completely by mutation of the RVXF motif. The PP1:MKLP1 interaction is only partially lost by mutation of this motif (Fig. S5C). This shows that MKLP1 either does not directly bind to PP1 (see previous point) or that MKLP1 has at least one other, more important PP1-binding motif.

All validated PP1-binding RVXF-motifs reside in an intrinsically disordered fragment of the polypeptide. Is the PP1-binding domain of MKLP1 predicted to be disordered? If not, the small reduction in (in)direct PP1:MKLP1 interaction by mutation of the VQF sequence may simply be explained by a conformational change of MKLP1 that partially disrupts its interaction with PP1 or a PP1-binding protein.

We are disconcerted that the Reviewer continues to dwell on our results indicating an interaction between MKLP1 and PP1 β even after the considerable changes and additions we made to our revised manuscript. We clearly stated that mutating the VQF sequence does not abolish the interaction between the two proteins but only reduces their affinity. We also do not believe that this is due to the presence of a second PP1-binding motif because we could not identify any other potential PP1-binding sites in MKLP1, and even if this site existed, this would not change the nature of our findings. The points we make in the paper are that: (1) this mutant has lower affinity for PP1 beta in yeast and *in vitro*; (2) it is de-phosphorylated less efficiently by PP1 beta *in vitro* (a result completely ignored by the Reviewer); and (3) it has a clear phenotype in cells that illustrates the importance of PP1 β -MKLP1 interaction (one of the important findings of this work; Fig. 7h-i). Together, these results convincingly demonstrate that the interaction between MKLP1 and PP1 β is genuine and of functional significance.

Finally, to answer the Reviewer's question, the IUPred2A software predicts that the MKLP1 C-terminal region after position 660 - which of course contains the VQF motif at 786-788 - is the largest intrinsically disordered region of this protein (see graphs below).

Reviewer 3

The revised manuscript satisfactorily addresses the points I raised in the first round of review. This is an interesting study and publication is recommended. However, a few additional minor points were noted during this second round of review. For the sake of precision and completeness of the manuscript, it is recommended that the following be addressed, although I do not need to approve these changes:

We are very pleased that the changes we introduced in the revised version of the manuscript satisfy the Reviewer, who now recommends publication. S/he only made some minor comments, which we have addressed in our point-by-point responses below.

1) Statistical significance of any observed differences (in frequency of multinucleated cells) from the control should be added to graphs in Figures 4H and 7I.

We have included statistical analyses and p values in the graphs in Figures 4h and 7i.

2) Figure 7I The graph legend (purple) is mislabelled- "MYPT1" instead of "MKLP1".

We apologize for this typo, which has now been corrected.

3) Pg 7 "...the levels of both mono(pS19)- and di(pT18 pS19)-phosphorylated MRLC levels were elevated in MYPT1 depleted cells (Fig 5a-b), which had also an abnormal cytoskeleton and numerous cortical blebs (Fig. 4F)." Legend of Figure 4. "Note that MYPT1 siRNA cells show abnormal cell and nuclear shape and disorganized microtubule and actomyosin cytoskeletal filaments."

The claimed cortical blebs are not evident from Figure 4F, maybe if actin had been stained, and only then could a claim about the actin filaments be made. It is suggested that this be re-worded.

We have now included arrowheads in Fig. 4f to mark the cortical blebs (which are actually visible even under simple brightfield phase contrast illumination). We stained with di(pT18 pS19)-phosphorylated MRLC in Fig. 4f to mark contractile actomyosin filaments. Staining for F-actin would visualize all actin filaments and not specifically actomyosin filaments. For these reasons, we think that our statement is correct and it does not need to be changed.

REVIEWERS' COMMENTS:

Reviewer #2 (Remarks to the Author):

None